# Predicting Emergent Abilities with Infinite Resolution Evaluation

**Shengding Hu**[1], **Xin Liu**[2], **Xu Han**[1,3]***Xinrong Zhang**[1], **Chaoqun He**[1], **Weilin Zhao**[1],
**Yankai Lin**[4], **Ning Ding**[1], **Zebin Ou**[5], **Guoyang Zeng**[6], **Zhiyuan Liu**[1]*, **Maosong Sun**[1]*
[1]Department of Computer Science and Technology, Tsinghua University
[2]Beijing Language and Culture University.
[3]Shanghai Artificial Intelligence Laboratory
[4]Renmin University of China.    [5]Zhihu Inc.    [6]Modelbest Inc.
hsd23@mails.tsinghua.edu.cn

## Abstract

The scientific scale-up of large language models (LLMs) necessitates a comprehensive understanding of their scaling properties. However, the existing literature on the scaling properties only yields an incomplete answer: optimization loss decreases predictably as the model size increases, in line with established scaling law; yet no scaling law for task has been established and the task performances are far from predictable during scaling. Task performances typically show minor gains on small models until they improve dramatically once models exceed a size threshold, exemplifying the "emergent abilities". In this study, we discover that small models, although they exhibit minor performance, demonstrate critical and consistent task performance improvements that are not captured by conventional evaluation strategies due to insufficient measurement resolution. To measure such improvements, we introduce PassUntil, an evaluation strategy with theoretically infinite resolution, through massive sampling in the decoding phase. With PassUntil, we conduct a quantitative investigation into the scaling law of task performance. The investigation contains two parts. Firstly, a strict *task scaling law* that is not conventionally known to exist, is identified, enhancing the predictability of task performances. Remarkably, we are able to predict the performance of the 2.4B model on code generation with merely 0.05% deviation before training starts, which is the first systematic attempt to verify predictable scaling proposed by GPT-4's report (OpenAI, 2023). Secondly, underpinned by PassUntil, we are able to study emergent abilities quantitatively. We identify a kind of **accelerated emergence** whose scaling curve cannot be fitted by standard scaling law function and has a increasing speed. We then examine two hypothesis and imply that the "multiple circuits hypothesis" might be responsible for the accelerated emergence.

*"See the world in a grain of sand"*

## 1 Introduction

Large Language Models (LLMs) (Devlin et al., 2018; Raffel et al., 2020; Brown et al., 2020; Chowdhery et al., 2022) have become a center of interest among AI researchers recently. These models, trained on expansive datasets and furnished with an enormous number of parameters, have demonstrated unparalleled proficiency across diverse domains, such as text generation (Dubois et al., 2023), code completion (Chen et al., 2021; Rozière et al., 2023), and academic test (Hendrycks et al., 2020).

The impressive success of these LLMs depends heavily on scaling up the model parameters and pre-training data volume. It has been consistently observed that, when considering a continuum of models with nearly identical architectures, larger models coupled with increased pre-training corpora consistently yield diminished training loss. This observation has been mathematically formalized as the scaling law of loss (Kaplan et al., 2020; Henighan et al., 2020), which states that the reducible loss achieved by the model in the log scale is linear to the model size in the log scale. Scaling law has provided guidance for the scientific scaling of LLMs, including determining the balance

---
*Corresponding Authors.

of the model size and pre-training data size (Hoffmann et al., 2022; Muennighoff et al., 2023). This has transformed what was once a somewhat blind scaling process into a methodology underpinned by empirical assurance. Nonetheless, such beneficial scaling law yield predictions solely on the loss, not extending to the real task performance encountered in practice. This divergence establishes a substantial gap in a comprehensive scaling-up methodology (Ganguli et al., 2022).

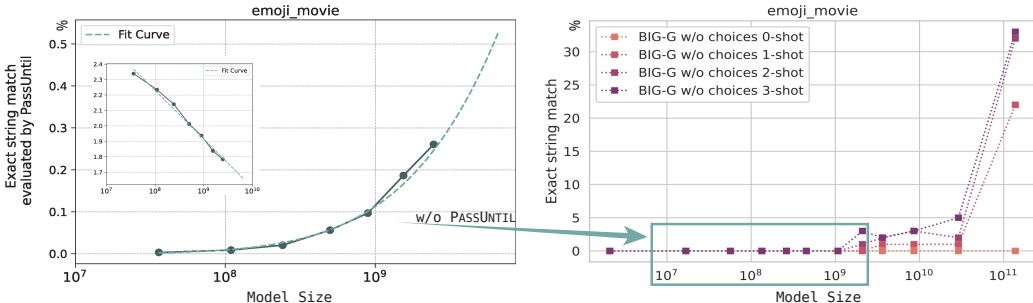

Figure 1: We can discriminate subtle performance improvement (left), which is evaluated as all zeros in conventional methods (right). The right figure directly uses Figure 9(a) in Sorscher et al. (2022) as a comparison, which the authors utilize to illustrate a "break-through" behavior in task performance. The internal figure inside the left figure shows the performances in a $\log(-\log(\cdot))$ space, which displays strong linearity, supporting the task scaling law (Eq.(3)).

The challenge in extending loss caling law to task performance predominantly stems from the *discontinuity* observed in task performance during scaling. Language models below a certain size yield trivial performance, i.e., random guessing on multiple choices or zero scores on generation tasks. However, when the model size surpasses a certain threshold, a distinct surge in performance appears, which leads to substantially non-trivial performance. This phenomenon is summarized as the "emergent abilities" (Srivastava et al., 2022; Wei et al., 2022a), and is observed across various model families and tasks. It seems that qualitative changes happen inside the model, which makes the model start to manifest unique capabilities. While these emerging phenomenon indicate that LLMs are becoming stronger, they complicate the prediction on task performance.

A pivotal question arises: **can we unlock predictable scaling of the task performance, from the apparent discontinuities?** We hypothesize that the perceived discontinuity from trivial to excellent performance might stem from limited evaluation resolution[1]. By employing a more nuanced resolution, one could potentially uncover the scaling law for tasks. The most related work to ours is Schaeffer et al. (2023), which proposes two methodology to make emergent abilities continuous, i.e., "change of metrics" and "increase resolution" by expanding test set size. Our motivation diverges from the "change of metric" approach of Schaeffer et al. (2023), which posits that employing other continuous metrics can cause emergent abilities to disappear. A limitation of alternative smooth metrics (e.g., distribution distance) is they yield insufficient insights into the target metrics (e.g., exact match) that evaluators intuitively perceive. In contrast, our method extends the "increase resolution" approach in a novel way, which target directly at predicting the performance such as code generation in our experiments.

We introduce an evaluation strategy named PASSUNTIL that, for the first time, enables quantitative exploration of the scaling properties of task performance. PASSUNTIL deploys extensive random sampling in the decoding phase (e.g., $10^5$ sampling times), and evaluates each sampling result *until* any generation *passes* the target test. Therefore, this evaluation strategy has infinite measurement resolution as long as computational resources are not bounded. Moreover, it can provide maximum likelihood estimates of target metrics such as accuracy and exact match. To refine our evaluation resolution and accuracy, we suggest fitting to instance-level scaling law since different test instances might have different speeds of performance improvement during scaling.

With the proposed evaluation strategy, we delve into the scaling law governing task performance. To begin with, we train two series of models ranging from 0.03B to 2.4B. These models strictly adhere to pre-training loss scaling law, providing a solid foundation for analyzing task performance scaling behavior. We mainly disclose two findings in our exploration.

---

[1]By "resolution", we view evaluation as a measurement of the real probability of completing a task. And resolution is the smallest probability difference that the evaluation strategy can detect.

Firstly, task performances are predictable with PASSUNTIL. We validate the presence of subtle but non-negligible performance in smaller models that can be captured by PASSUNTIL. These performances are on the order of $10^{-5}$ and exhibit steady enhancement as the model scales up. Subsequently, we derive the mathematical form of **task scaling law**, experimentally verifying an almost strict linear relationship between $\log(-\log(\text{PU}))$ and $\log(N)$, where PU denotes the estimation of target metric given by PASSUNTIL and $N$ is the number of model parameters. This relationship enables us to attain highly accurate predictions. For instance, in the code generation task, our predictions exhibit a mere 0.05% deviation from the actual values.

Secondly, we discover a phenomenon of **accelerated emergence**. To begin with, we discover that the shape of the task scaling curve is not uniform across tasks. Several task manifest scaling functions that diverge from the typical task scaling law. In other words, their scaling curve is smooth and incremental but can not be fitted by the typical scaling law function. Their scaling curve of $\log(-\log(\text{PU}))$ w.r.t. $\log(N)$ is concave, which is akin to an acceleration in the performance scaling speed. We provide a mathematical definition of such phenomenon. With the quantitative definition, we exclude a possible multi-step reasoning explanation (Schaeffer et al., 2023), and propose an alternative hypothesis. This hypothesis is predicated on potential transformer circuits (Nelson et al., 2021) that are used to explain the "grokking" phenomenon (Power et al., 2022; Varma et al., 2023). It is in harmony with the observed scaling function.

Our work represents the first open-source attempt regarding the predictability of task performance. While GPT-4's report (OpenAI, 2023) has initiated this exploration, it has not provided comprehensive details. We will open-source all checkpoints to facilitate future research in this direction.

## 2 RELATED WORK

Predicting task performance before training is an aspirational objective for the development of predictable AI systems, and a multitude of studies approach this aim from various perspectives.

**Loss Scaling Law.** Scaling phenomena have been observed across a broad spectrum of deep learning architectures. The power-law scaling behavior of loss in RNN-based models is investigated in Hestness et al. (2017). Kaplan et al. (2020) delineate the loss scaling trends for Transformer-based language models and explores the scaling behavior of optimal hyper-parameters. They formally established the following scaling law

$$L = cN^{-\alpha} + L_0, \tag{1}$$

where $N$ is the number of non-embedding parameters of LLM, $c, \alpha$ are positive coefficients, and $L_0$ is the irreducible loss representing the randomness in data. This formulation has catalyzed the proliferation of LLMs. Subsequently, scaling laws are established for various domains and scenarios, including multi-modality (Henighan et al., 2020; Zhai et al., 2022), computation constraint scenario (Hoffmann et al., 2022), data engineering (Muennighoff et al., 2023; Sorscher et al., 2022), and reinforcement learning (Gao et al., 2023). Yao & Wang (2023) extend the scaling law into loss prediction by introducing hyper-parameter scaling methods. The relationship of our work with these existing literature is twofold. First, these works concentrate on training and validation loss metrics, which do not reliably predict task performance. Second, our research builds on these scaling laws and extends the mathematical form of Eq.(1) to the scaling law of task performance.

**Scaling Behavior of Task Performance.** Despite the predictable decrement in LLM loss, task performance improvements are twisted during scaling. While some tasks, predominantly those relying on memorization of knowledge, have shown progressive improvement, numerous tasks exhibit breakthrough behavior as model size increases (Srivastava et al., 2022; Wei et al., 2022a). Wei et al. (2022a) illustrate that the concept of "emergence" is also pertinent to prompting techniques such as Chain-of-Thought (Wei et al., 2022b) and In-context Learning (Brown et al., 2020), complicating the pursuit of understanding the scaling law of task performance. It appears that the law of loss scaling offers no assurance for task performance, engendering a lack of guidance in pre-training methodology. Fortunately, several studies endeavor to demystify these emergent abilities. GPT-4's technical report (OpenAI, 2023) reports that GPT-4's task performance can be predicted with less than $1/10000$ of computation, albeit without disclosing the methodology and acknowledging that certain abilities are still beyond prediction. Subsequent research (Schaeffer et al., 2023) attributes emergence to two reasons. The first one is non-smooth metrics. We disagree with it since the alternative metrics could not explain the sudden increase in target metrics such

as exact match, which are of paramount interest to us. We align with their second attribution to improve resolution by adding more test samples. Different from their method, we propose a practical method to improve resolution without the need of adding test samples. Our work is also the first open-source attempt to quantitatively investigate the scaling behavior of task performance, proposing task scaling law and accelerated emergence phenomenon.

## 3 PILOT EXPERIMENTS ON INCREASING RANDOM SAMPLE NUMBERS

We initiate our exploration by visualizing the effect of improving evaluation resolution on open-sourced models. We choose four small models and evaluate them on two subsets of BigBench task (Srivastava et al., 2022): Emoji Movie and Date Understanding (see Appendix D.4.2 and D.4.3 for the subsets). We employ beam search and random sampling (with three sample times: 1, 100, and 10,000) during decoding. If any sampled answer of a test instance is evaluated as correct, then the instance is marked as "passed". We present the number of passed instances in Figure 2.

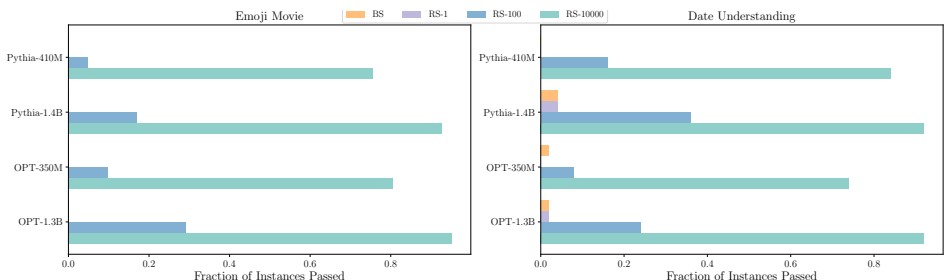

Figure 2: BS denotes beam search, RS-$K$ denotes random sampling $K$ times.

We can see that even for such tasks presenting substantial difficulty to small models, most instances are passable with enough random sampling times, which will contribute to the subtle task performance improvement. Inspired by this observation, we propose our evaluation strategy that centered around improving the resolution of evaluation.

## 4 METHODS

In this section, we describe our methods to increase the resolution of evaluation, which empowers the investigation of the scaling behavior of task performance. The first is an evaluation strategy PASSUNTIL, and the second is an instance-level scaling curve fit. We also derive the task scaling law based on the loss scaling law.

### 4.1 INFINITE RESOLUTION WITH PASSUNTIL

We view task performance evaluation as the measurement of the probability of a model passing [2] a task. Given a task instance $s$, suppose the probability that a model pass it is $P(s)$, our job is to estimate $\mathbb{E}_s[P(s)]$. Randomly sampling a fixed time $K$ could estimate $P(s)$. However, it is hard to define the budget $K$ that is both acceptable in computation and has enough resolution for hard samples that have small $P(s)$. We propose PASSUNTIL, which performs an evaluation right after an answer is generated and determines whether it is passed before we sample the next generation. We stop sampling until $r$ (a constant) samples have passed the evaluation and record the sampling number $K$. We name the estimate of $P(s)$ as the PASSUNTIL score PU, which is defined as

$$\text{PU} = \frac{r}{K} \tag{2}$$

Theoretically, PU has the capability to measure success rates that are infinitesimally small. The PASSUNTIL has the following properties.

---

[2]The definition of "pass" does not need to be generating exactly the ground truth answer. For example, suppose we predict model's performance on AlpacaEval (Li et al., 2023b), we can define "pass" as the model generation being better than GPT-4, judged by GPT-4. Therefore the "pass" has broad application.

**Theorem 1.** PU *is a maximum likelihood estimate for* $P(s)$.

*Proof.* The failure time $f = K - r$ follows the negative binomial distribution with success probability $P(s)$. $r/K$ is known to be an maximum likelihood estimate for $P(s)$. $\square$

In practice, we set $r$ to as small as 1 or 2 considering the efficiency of evaluation. We also set the upper bound of $K$ to a large number, such as $10^5$, to prevent endless sampling if we encounder an extremely low $P(s)$. Note that many instances stop before reaching this upper-bound. Next we discuss the necessity and limitations of PASSUNTIL.

**Necessity.** Generally, deriving $P(s)$ theoretically from the token probability on the ground truth solution is not feasible. This is due to two primary facts: firstly, there are likely to be multiple viable solutions; secondly, even though there is only one solution, there exist multiple decoding approaches besides the optimal tokenization to decode the solution[3].

**Limitations.** (1) Currently, our evaluation strategy is designed to be applicable when a random baseline achieves $P(s) = 0$. In the context of multiple-choice grade as the evaluation metric, evaluations tend to exhibit a biased high score relative to the true performance of the model (e.g., $P(s) = 0.25$ with random guess for four options). This random noise can overshadow the improvements made by smaller models. The exploration of scaling law for tasks with non-zero random baselines remains a subject for future research. (2) We currently only consider random sampling as a viable target decoding strategy due to its widespread use in LLMs. Using beam search as target decoding strategies and their relationship with random sampling poses an interesting avenue for future exploration and study.

## 4.2 FROM LOSS-SCALING LAW TO TASK SCALING LAW

Then, we derive the task scaling law that PASSUNTIL will follow. We assume that the test loss of generating the next token decreases according to the scaling law of Eq.(1).

$$\text{PU} \sim \prod_{i=1}^{|y|} P(y_i|x_{1:|x|}, y_{1:i-1}) = \prod_{i=1}^{|y|} \exp(-c_i N^{-\alpha_i} - L_{0i}), \tag{3}$$

where $x_{1:|x|}$ is the input sequence and $y_{1:|y|}$ is the most probable sequence that decodes the correct answer (assuming its dominance compared to other sequences). Assume that the test sample is passable given a sufficiently potent LLM, then the irreducible loss for each token $L_{0i}$ approaches 0. And assume the test loss of each token in the answer is decreasing with uniform speed when scaling (i.e., $a_i = a, \forall i$), we can derive the following function for PU on task performance:

$$\text{PU}(c, \alpha; N) \sim \exp(\sum_i -c_i N^{-\alpha}) = \exp(-cN^{-\alpha}) \tag{4}$$

where $c = \sum_i c_i$. The resulting mathematical model is similar to that in GPT-4 technical report (OpenAI, 2023) and Equation (4) in Schaeffer et al. (2023).

## 4.3 FITTING STRATEGY

**Dataset-level Fit.** When fitting the parameters $c, \alpha$ in PU, a dataset-level fit is plausible. For the $j$-th model in the scaling curve, the individual test sample's PU is first averaged over the test set to procure $\log(-\log(\text{PU}(N_j)))$, followed by a linear regression to $\log N_j$.

**Instance-level Fit.** We notice that differences between instances lead to different scaling behaviors, which means a dataset-level fit might not be accurate when the difficulty in the test set is diverse. For example, PU on easy questions get saturated to 1 on a small model while the hard questions still receive trivial performance (see Appendix B.1 for illustration). We propose to fit an individual PASSUNTIL score (IPU) for each question and aggregate them into an estimate for the whole dataset.

$$\text{PU}(\{c_s, a_s\}; N) = \frac{1}{|S|} \sum_s \text{IPU}(c_s, a_s; N) \tag{5}$$

---

[3]For example, [4513], [717,18], and [16,17,18] all decode into string "123" in GPT-4's tokenizer with vocab "cl100k-base".

## 5 PREDICTABLE SCALING EXPERIMENTS

In this section, we demonstrate how the proposed framework works in practice. We first pre-train two series of language models ranging from 0.03B to 2.4B using two dataset mixtures. We predict the performance of the 2.4B model based on the performance of the rest of the models in the series.

### 5.1 SCALING CONFIGURATIONS.

**Model Configurations.** We propose to keep a consistent "shape" of the Transformers while expanding their sizes. For the $i$-th model in the scaling curve, we set the number of layers to be $4i$, the number of attention heads to be $\lfloor \frac{i(8+i)}{4} \rfloor$, and the dimension of head to be $64$. This results in the hidden state's dimension $d_m$ being $d_h n_h$. We set the dimension of the feed-forward layer to be $2.5d_m$. The specific values are listed in the model configurations in Table 3 of Appendix D.1. The architecture is similar to LLaMA (Touvron et al., 2023a) (see Appendix D.1 for details).

**Pre-training Corpora.** For series 1, we use the StarCoder dataset (Li et al., 2023a) as our pre-training data. For series 2, we use a mixture of StarCoder and Pile (Gao et al., 2020) dataset. Leveraging the optimal compute LLMs (Hoffmann et al., 2022), we set the maximum pre-training tokens for each model size to be the $20N$, where $N$ is the number of non-embedding parameters of the model. The detailed portion within the data mixture can be seen in Appendix D.2.

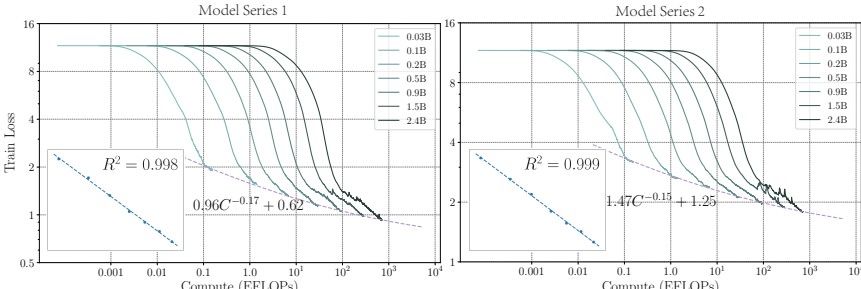

Figure 3: Training loss of the two series of models trained on different data mixtures. The internal figure illustrates the end-step reducible loss relative to model size, represented in logarithmic scale.

**Hyper-parameters.** Hyper-parameters are also of paramount importance in training a series of models that scale successfully. We examine the cosine learning rate scheduler, aligning our approach with that of Hoffmann et al. (2022), and determine the critical batch size in accordance with Kaplan et al. (2020). Nonetheless, due to constraints in space, we move the details to Appendix D.3.

### 5.2 LOSS SCALING LAW VERIFICATION.

We present the training loss curves for models in Figure 3. It is evident that the end-step training losses decrease in line with the scaling law. These empirically observed loss scaling laws lay a foundation for the subsequent approximation of task performance. Note that despite the occurrence of the loss spike in the 1.5B and 2.4B models, convergence to the scaling law is ultimately achieved, exemplifying the robustness of such an empirical law.

### 5.3 DATASET-LEVEL FIT

We select HumanEval (Chen et al., 2021), Emoji Movie, and Date Understanding (Srivastava et al., 2022) as the evaluation tasks. Note that Emoji Movie is conventionally cited as representing "emergent abilities" (Srivastava et al., 2022) (see the right figure in Figure 1). HumanEval is assessed using a zero-shot learning setting, while Emoji Movie and Date Understanding are evaluated employing 4-shot In-context Learning (Brown et al., 2020). We additionally use Chain-of-Thought Reasoning (Wei et al., 2022b) for Emoji Movie. See Appendix D.4 for the illustration and evaluation details of each task. We remove the distracting test instances from our evaluation list. For Emoji Movie, we remove the movie names that are common words (e.g., "*it*") identified by NLTK (Bird et al., 2009). These common words make the exact string match susceptible to random guess's correctness ( See Appendix D.5 for details).

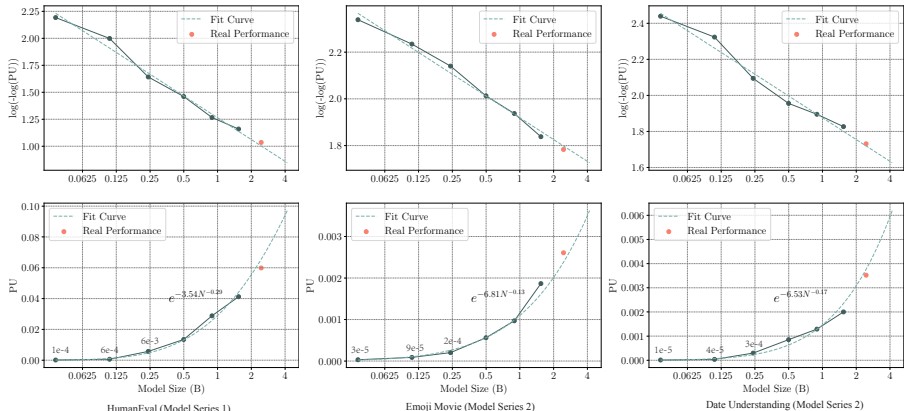

Figure 4: Task performance scales predictably with model scale. The red points denote the real performance of 2.4B model, which are close to the task scaling laws fitted from 0.03B to 1.5B.

We observe that all three tasks exhibit a strong linear relationship between $\log(-\log(\text{PU}))$ and $\log(N)$, verifying the success of task scaling law given by Eq.(3). The estimation of the scaling law functions utilizes the 0.03b to 1.5B models, which predicts the performance of the 2.4B model with small yet acceptable deviations.

## 5.4 INSTANCE-LEVEL FIT

According to § 4.3, we take the difference among test samples into consideration to improve the estimation. We plot how instance-level PASSUNTIL scales in Figure 13 of Appendix E.4. The fitted curves demonstrate that the performances of different instances not only originate from unique starting points but also scale at varying speeds. Nevertheless, they can be fitted by task scaling law individually. Some instances deviate from the scaling law, which needs future investigation.

| Method | HumanEval (1) | HumanEval (2) | Date Understanding (2) | Emoji Movie (2) |
|---|---|---|---|---|
| Real Value | 0.05990 | 0.04279 | 0.00346 | 0.002608 |
| Dataset-level Fit | 0.06550 | 0.05191 | 0.00377 | **0.002381** |
| Instance-level Fit | **0.05987** | **0.04402** | **0.00352** | 0.003112 |

Table 1: Prediction of our framework compared to the real performance on two series of models. The number after the task denotes the model series used in the evaluation.

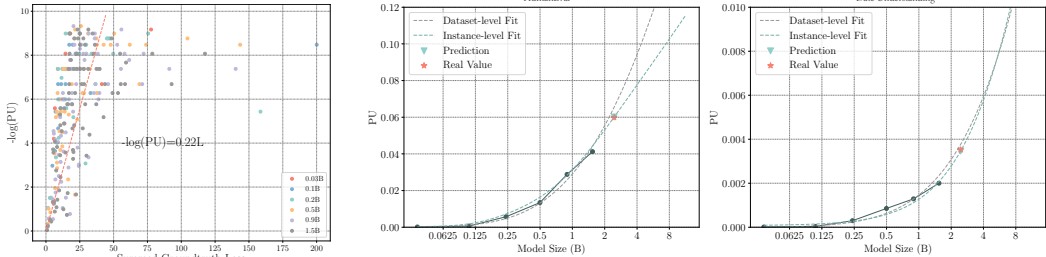

Figure 5: PU w.r.t. the test loss on HumanEval of model series 1.

Figure 6: We successfully predicted the performance of 2.4B model with 0.05% deviation (left) and 1.7% deviation (right).

**Estimating PASSUNTIL from Test Loss.** Estimating at the instance level presents challenges for hard instances that lack adequate non-zero PU values for fitting. These samples may also contribute to PU as the model size increases. We suggest leveraging test loss on ground truth answers to assist the prediction for such instances (See Appendix A.2 for a detailed discussion of its validity). We leverage the "easy" instances, which have both test loss and non-zero PU to estimate the relation between test loss and PU (Figure 5). Then we predict the test loss of each instance on 2.4B model based on 0.03B ∼ 1.5B models. Finally, we transform the predicted test loss to predicted PU according to the aforementioned relationship. Details are presented in Appendix E.2. We provide the final prediction result of 2.4B model in Table 1, and draw the predicted PU curve in Figure 6. We can see that the predictions are accurate, with only 0.05% difference on HumanEval of series 1 and 1.7% difference on Date Understanding of series 2.

## 6 QUANTITATIVE ANALYSIS OF EMERGENCE

Building on the discovery of the predictability of task performance, we proceed with our investigation into a quantitative analysis of scaling behavior of broader range of tasks. We prove that even with the refined resolution brought by PASSUNTIL and predictability of other emergent abilities, there are still certain abilities hard to be predicted. We establish their mathematical definitions, and examine the possible explanations for such scaling behaviors.

We study the scaling curve on the "Unnatural In-context Learning (UICL)" categories in Big-Bench (Srivastava et al., 2022). "Unnatural In-context Learning" is a set of 8 tasks designed to specifically study the in-context learning ability. These tasks involve input-output pairs that have been intentionally altered to deviate from the typical training distribution, thereby necessitating the model's focus on unconventional in-context patterns. Task details and examples are in Appendix D.4.4. We randomly select 20 questions in the test set from each task and sample 4-shot examples from the remaining questions to serve as in-context examples. The evaluation metric employed is the exact match, and the upper bound sampling time is set to $10^5$. When fitting the scaling curve, we only utilize the dataset-level PASSUNTIL since these test instances are manually constructed to test one skill of LLM and thus might be devoid of difficulty variation. Since our test set is small, we bootstrap 100 times from the 20 question's test result and use the bootstrapped to calculate the standard error of each PASSUNTIL estimate (shown in the green hue in the Figures).

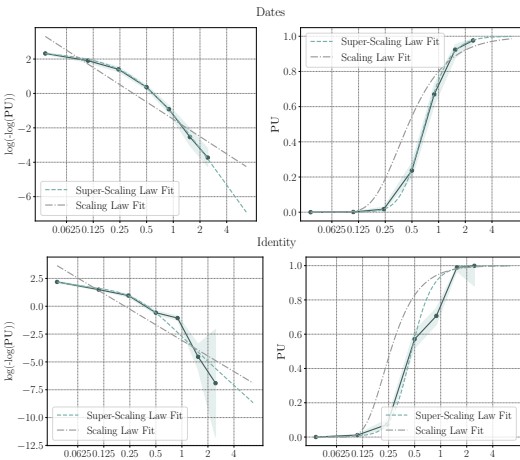

Figure 7: Scaling curve for task "Dates" and "Identity". Concave functions are observed between $\log(-\log(\text{PU}))$ and $\log N$. Scaling law fit curves are in grey and super-scaling law fit curves are in green.

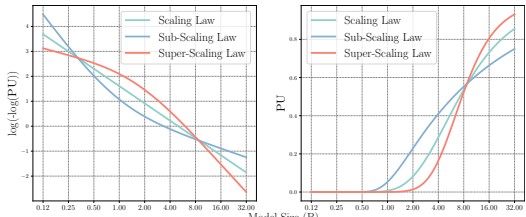

Figure 8: Three basic types of scaling curve, corresponding to convex, linear, and concave function between $\log(-\log(\text{PU}))$ and $\log N$.

**Categorization of Emergence.** The evaluation on task "Dates" and "Identity" is shown in Figure 7. Other tasks are shown in Appendix E.3. "Dates" exhibit very smooth and consistent improvement starting from 0.03B, while the other tasks are a bit twisty. Nevertheless, 5/8 of these in-context learning tasks display a strictly concave function between $\log(-\log(\text{PU}))$ and $\log N$. The others (3/8) miss 1 or 2 valid estimation points due to their extreme difficulty for 0.03B and 0.1B models, since 0 PASSUNTIL is obversed even with $10^5$ sampling time, which we left for future exploration. The 5/8 tasks deviates from the scaling law (Eq.(3)) which requires this function to be linear. This means, unlike those tasks governed by the task scaling law, where "growth speed" $\alpha$ is uniform across different model sizes, **there exist some tasks that see an increase in "growth speed" $\alpha$ as models enlarge. This phenomenon exemplifies an accelerated emergence phenomenon.** To provide concrete discussion of accelerated emergence, we provide our categorization of task scaling curves first.

**Mathematical Definition of Emergence.** Since the loss scaling law of Eq.(1) is the only widely accepted principle during model scaling, we rely on its derived task scaling law of Eq.(3) as a separator between emergence and other scaling behavior.

**Definition 1.** *Given a spectrum of models, we let the number of non-embedding parameters be variable $N$, suppose the $\text{PU}(N)$ estimated by PASSUNTIL on a task is a continuous function of $N$. Define $F(N) = \log(-\log(\text{PU}(N)))$, then the scaling curve of a task can be categorized into three basic main categories [4]:*

---

[4] if $F(N)$ has both convex and concave parts, then we can call it mixed growth.

1. *if $F(N)$ is a linear function of $\log N$, then the task obeys scaling law growth.*

2. *if $F(N)$ is a convex function of $\log N$, then the task obeys sub-scaling law growth.*

3. *if $F(N)$ is a concave function of $\log N$, then the task obeys super-scaling law growth, or "accelerated emergence".*

Figure 8 shows visualizations of three types of growth. Qualitatively, the scaling curves of all three types appear analogous to exponential growth when performance starts to become noticeable. However, they are qualitatively different. Task scaling curves with task scaling law growth or sub-scaling law growth are easier to predict and control, whereas accelerated emergence is not easy to predict, which might go out of control when the model gets larger.

**Cause of Shape of Scaling Curve.** The above mathematical definition provides us the opportunity to examine the hypothesis regarding the genesis of these scaling behavior. Here, we first study the following hypothesis: Emergent abilities may be induced by multi-step reasoning (Srivastava et al., 2022; Wei et al., 2022a; Schaeffer et al., 2023).

We prove that, surprisingly, **"multi-step reasoning" leads to sub-scaling law growth**.

**Theorem 2.** *Suppose each reasoning step's success rate, measured by PASSUNTIL obeys the scaling law growth, then the multi-step success rate follows the sub-scaling law growth.*

*Proof.* Suppose the success rate of reasoning step $i$ obeys a scaling law growth with coefficient $c_i$ and $\alpha_i$, then $F(N) = \log\left(\sum_i c_i \exp\left(-\alpha_i \log N\right)\right)$. Using Cauchy–Schwarz inequality, we can prove that $\frac{\partial^2 F}{\partial(\log N)^2} \geq 0$. Therefore, the scaling curve is convex. See Appendix C.1 for more. □

This proof can also be understood more intuitively: the growth speed will initially be boosted by the improvement of those easy steps, and eventually be bounded by the most difficult steps, thus showing a decreasing growth speed. Then, we propose an alternative hypothesis: suggesting that multiple neural "circuits" (Nelson et al., 2021) may be represented within the LLMs, and that as long as one such circuit can successfully solve the test instance, the test instance is deemed passed. This hypothesis is inspired by the explanation of "grokking" phenomenon given by Varma et al. (2023). They propose that there exists a memorization circuit and a generalization circuit inside the transformers, and the "grokking" phenomenon is led by the generalization circuit getting more efficient than the memorization circuit during training. We will demonstrate that with this hypothesis, the scaling curve exhibits characteristics of emergence.

**Theorem 3.** *Suppose multiple circuits $i$ exist in the LLMs that are responsible for solving the task, and each displays scaling law growth and has $\mathrm{PU}_i$. And suppose the success rate of the task is the majority voting of these circuits, i.e., $F(N) = \log\left(-\log\max_i \mathrm{PU}_i\right)$. Then, $F(N)$ is a concave function of $\log N$.*

*Proof.* $F(N) = \min_i(\log c_i - \alpha_i \log N)$. Since the minimum operator keeps concavity, $F(N)$ is a concave function of $\log N$. See Appendix C.1 for a more elaborated proof. □

We loosely test the hypothesis by fitting the scaling curve for the UICL task. In practice, similar to Varma et al. (2023), we adopt a soft version of the majority voting. We apply a weighted combination between two circuits. And we assume the number of the circuits is 2. Therefore, we fit $w_1(\alpha_1 \log N - \log c_1) + w_2(\alpha_2 \log N - \log c_2)$ to $F(N)$, where $w_1$ and $w_2$ is given by the Softmax of $\alpha_i \log N - \log c_i$. The resulting fit curve is demonstrated in the green line in Figure 7 and Appendix E.3. We can see that this hypothesis produces fit curves that align more accurately with the observed performance scaling curve.

## 7 CONCLUSION.

Our work introduces a novel evaluation strategy capable of detecting minimal performance improvements during model scaling, thus opening avenues for quantitatively measuring the task scaling laws and the emergence abilities. This method has enabled the successful prediction of the task performance of larger models. Additionally, we have performed a quantitative analysis of emergent abilities, providing a clearer insight into their nature and origination. This research not only enhances our understanding of LLMs' scaling properties but also sets the stage for future explorations in scientific scale-up of LLMs.

ETHICAL STATEMENT

In this paper, we demonstrate that although we can predict a set of emergent abilities, the accelerated emergence remains hard to be predicted. The hypothesis regarding the cause of accelerated emergence implies that we need a better understanding of the working mechanism to produce accurate predictions for such emergent ability. Without an understanding of the working mechanism, any fit curve to the early stage of task performance improvement might be governed by another stronger, yet unknown, "generalization" circuit when the model gets sufficiently large. Thus, this hypothesis calls for deeper research into the mechanism of LLMs to prevent the safety concerns brought by accelerated emergent abilities.

REPRODUCIBILITY STATEMENT

We will open-source and all evaluation scripts for reference.

ACKNOWLEDGEMENTS

This work is supported by the National Key R&D Program of China (No.2022ZD0160501).

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

**Note: clicking each 👆 in the appendix will allow you to jump back to the corresponding position in the main paper to continue reading.**

# A    DISCUSSION

## A.1    LIMITATIONS

Our work has several limitations.

1. **Scale Limitation.** Firstly, we currently do not extend the prediction of task performance to much larger models (e.g., 10B and more). We will try to scale up the experiment in the future.

2. **Scope Limitation.** Secondly, we are not claiming that we can accurately predict the task performance on all tasks. For example, we only *fit* the scaling curve for the tasks that display emergence. We still have a long way to go before we can *predict* these tasks. Even for the tasks that might not display "emergence", we currently do not complete a thorough prediction for them. We will add predictions on more of these tasks in the future. That said, predictable scaling, as OpenAI points out (OpenAI, 2023), is still a very challenging and aspirational goal for AI researchers. Our work serves as the initial attempt to it.

3. **Explanation Limitation.** Thirdly, although we propose a hypothesis regarding the cause of accelerated emergence, our validation for the hypothesis is superficial. We satisfactorily fit the scaling curve under this hypothesis. However, whether this hypothesis is true from the underlying mechanism remains unknown.

## A.2    DISCUSS OF THE USE OF LOSS AS AN ASSISTANCE METRIC

In our experiments of Individual PASSUNTIL, we use loss on ground truth as an assistance to PASSUNTIL, which may raise a misunderstanding: why don't you directly use loss to predict the performance? We provide a detailed illustration below.

1. It's important to distinguish between "loss is not predictive of task performance" and "loss can help predict task performance." The former suggests that loss is a not sufficient statistic for estimating task performance without other measurement, while the latter indicates that loss is one of useful factors in improving prediction accuracy. In our paper, we clearly verify both statements. Without utilizing the PassUntil method, one cannot deduce actual performance (accuracy) solely from loss values. For example, a loss of 1.0 does not directly translate to an accuracy of 0.2 for a task. And actual performance must be empirically measured. Furthermore, as shown in Figure 5, the loss of an individual sample does not have a one-to-one correlation with PassUntil results, much less with discrete accuracy.

2. However, loss does provide useful information. Once we measure PassUntil across a large sample set, we can establish a statistical relationship between loss and PassUntil (not possible if we only rely on loss data). This relationship can enhance our prediction accuracy.

3. The incorporation of loss for improved predictions is driven by practical considerations, such as limited computational resources, rather than being a necessity. Figure 4 demonstrates that even without loss data, we can accurately predict task performance. Imagine a scenario where we can measure every sample with sufficient resolution to ensure each is passed at least once; in such a case, loss data would not be necessary.

# B    SUPPLEMENTARY MATERIALS FOR PASSUNTIL

In this section, we provide some additional comments about our evaluation strategy. We present our intuition for instance-level PASSUNTIL.

## B.1    INSTANCE-LEVEL PASSUNTIL INTUITION.

👆 Table 2 delineates the PASSUNTIL for both an easy and a challenging instance within HumanEval. It was observed that with an increase in model size, the easier instance (index 24) exhib-

ited a higher PU. However, the more challenging instance (index 20) continued to manifest trivial performance, suggesting a potential variance in their respective scaling curves. Blindly averaging performance over instances will make the improvement on hard instances vanish compared to the easy ones, leading to an inaccurate prediction after the model gets saturated in the easy instances.

| Instance index | PASSUNTIL | | | | | |
|---|---|---|---|---|---|---|
| | **0.03B** | **0.1B** | **0.2B** | **0.5B** | **0.9B** | **1.5B** |
| 20 | 0 | 0 | 0 | 0.000625 | 0.001875 | 0.008125 |
| 24 | 0.00375 | 0.05125 | 0.350625 | 0.3625 | 0.568125 | 0.796875 |

Table 2: In HumanEval, an easy instance (index 24) gets a much higher PU compared to the hard one (index 20).

## C  SUPPLEMENTARY MATERIALS ON EMERGENT ABILITIES

### C.1  THEORETICAL ANALYSIS OF HYPOTHESIS

👆 We present the proof of two theorems about the cause of emergent abilities in Section 6 briefly. In this section, we provide the elaborated proofs.

**Theorem 2.** *Suppose the success rate of each reasoning step $i$, measured by* PASSUNTIL, *obeys the scaling law growth. Then the multi-step's success rate follows the sub-scaling law growth.*

*Proof.* Suppose the PU of reasoning step $i$ obeys a scaling law growth with coefficient $c_i$ and $\alpha_i$, The overall success rate is

$$
\begin{aligned}
F(N) &= \log\left(-\log\prod_i P_i\right) \\
&= \log\left(-\log\prod_i \exp\left(-c_i \exp(-\alpha_i \log N)\right)\right) \\
&= \log\left(\sum_i c_i \exp\left(-\alpha_i \log N\right)\right)
\end{aligned}
\tag{6}
$$

Then we take the second derivative of the $F(N)$ over $\log N$, we can get

$$
\begin{aligned}
\frac{\partial^2 F}{\partial (\log N)^2} = &\frac{\sum_i \alpha_i^2 c_i \exp(-\alpha_i \log N)\sum_i c_i \exp(-\alpha_i \log N)}{(\sum_i c_i \exp(-\alpha_i \log N))^2} \\
&- \frac{(\sum_i \alpha_i c_i \exp(-\alpha_i \log N))^2}{(\sum_i c_i \exp(-\alpha_i \log N))^2}
\end{aligned}
\tag{7}
$$

Let $k_i = c_i \exp(-\alpha_i \log N) > 0$, the Eq.(7) is

$$
\frac{\sum_i \alpha_i^2 k_i \sum_i k_i - (\sum_i \alpha_i k_i)^2}{(\sum_i k_i)^2}
\tag{8}
$$

Using Cauchy–Schwarz inequality, we can prove that

$$
\frac{\partial^2 F}{\partial (\log N)^2} \geq 0, \quad \forall \alpha_i > 0, c_i > 0
\tag{9}
$$

Only when $\alpha_i \sqrt{k_i}/\sqrt{k_i} = \text{Constant}$, the equation holds, i.e., when all the steps in the reasoning chain scale with the same speed. Thus, $F(N)$ is a convex function of $\log N$, and the scaling curve exhibits sub-scaling law growth. □

**Theorem 3.** *Suppose multiple circuits exist in the LLMs that are responsible for solving the task, each displays scaling law growth, the* PASSUNTIL *of the task is the majority voting of these circuits, i.e., $F(N) = \log\left(-\log\max_i P_i\right)$ Then, $F(N)$ is a concave function of $\log N$.*

*Proof.*

$$F(N) = \log\left(-\log\max_i \exp\left(-c_i \exp(-\alpha_i \log N)\right)\right)$$
$$= \log\min_i c_i \exp(-\alpha_i \log N) \tag{10}$$
$$= \min_i(\log c_i - \alpha_i \log N)$$

Since the minimum operator keeps concavity, $F(N)$ is a concave function of $\log N$. □

## D DETAILS OF EXPERIMENTAL CONFIGURATIONS

In this section, we detail the model configurations, training configurations, and data mixtures used for the two series of models.

### D.1 MODEL CONFIGURATION

👆 Table 3 shows the detailed model configurations and training configuration of the series models in the scaling curve, which aims to keep a uniform "shape" while expanding the model size. We use a similar architecture to Llama 2 (Touvron et al., 2023b). Some minimal differences include: we use tied embedding between the input and output embeddings, and we use gated-GeLU (Hendrycks & Gimpel, 2016) instead of gated-SiLU (Shazeer, 2020).

| Name | i | N (B) | $d_m$ | $d_{ff}$ | $d_h$ | $n_h$ | $L$ | BS (M) | TS | Tokens (B) |
|------|---|-------|-------|----------|-------|-------|-----|--------|-----|-----------|
| \ | i | \ | $d_h n_h$ | $2.5d_m$ | 64 | $\lfloor\frac{i(8+i)}{4}\rfloor$ | $\lfloor 4i\rfloor$ | \ | \ | \ |
| 0.03B | 3 | 0.036 | 512 | 1280 | 64 | 8 | 12 | 0.33 | 2196 | 0.72 |
| 0.1B | 4 | 0.109 | 768 | 1920 | 64 | 12 | 16 | 0.88 | 2464 | 2.18 |
| 0.2B | 5 | 0.241 | 1024 | 2560 | 64 | 16 | 20 | 1.57 | 3064 | 4.82 |
| 0.5B | 6 | 0.499 | 1344 | 3360 | 64 | 21 | 24 | 2.10 | 4758 | 9.99 |
| 0.9B | 7 | 0.892 | 1664 | 4160 | 64 | 26 | 28 | 2.95 | 6049 | 17.9 |
| 1.5B | 8 | 1.542 | 2048 | 5120 | 64 | 32 | 32 | 4.26 | 7230 | 30.8 |
| 2.4B | 9 | 2.45 | 2432 | 6080 | 64 | 38 | 36 | 5.51 | 8900 | 49.0 |

Table 3: Model configurations and training configurations of the models in the scaling curve. N(B) represents the number of non-embedding parameters of the model, measured in billions. BS(M) indicates the number of tokens in a batch (i.e., batch size) used to train the model, measured in millions. TS denotes the training steps. Tokens(B) refers the total number of tokens used to train the model.

### D.2 PRE-TRAINING CORPORA

👆 We pre-train two series of LLMs using different data mixtures to demonstrate the generality of our experiments. Tables 4 and 5 respectively display the specific data mixture proportions for Series 1 and Series 2 LLMs.

### D.3 HYPER-PARAMETERS STUDY

👆 **Learning Rate.** We use a cosine learning rate scheduler, analogous to those in preceding studies (Touvron et al., 2023a;b; Hoffmann et al., 2022). The maximum learning rate is consistently fixed at 0.01 across varying model scales, with no significant loss explosion at this rate. This stability is potentially attributed to our normalization strategies (Yang et al., 2022) and increased batch size across scales. Echoing findings from Hoffmann et al. (2022), we ascertain that for training LLMs up to a specific end step, the optimal cycle length of the cosine learning rate scheduler is equivalent to the end step. Deviations from this optimal cycle length, either longer or shorter, result in sub-optimal performance.

**Batch Size.** To estimate the optimal batch size required for model pre-training, we replicate the experiments in alignment with Kaplan et al. (2020) to determine the optimal batch size of a model

and adjust the real batch size slightly from the optimal batch size to maximize GPU utility. The values of batch sizes and train steps are listed in Table 3.

### D.4 TEST SET CONFIGURATIONS

☝ In this section, we introduce the test sets and evaluation details in our experiments.

#### D.4.1 HUMANEVAL

The HumanEval (Chen et al., 2021) dataset released by OpenAI encompasses 164 programming problems. Each problem is composed of a function signature, a docstring, a body, and multiple unit tests. Our assessment of this dataset is conducted utilizing a zero-shot approach. The completion of code, as generated by LLMs, is deemed passed only if it successfully passes all unit tests. For our evaluations, we set the upper bound of sampling times in PASSUNTIL to $10^4$.

#### D.4.2 EMOJI MOVIE

☝ Emoji Movie is a subtask of BigBench (Srivastava et al., 2022) and requires LLMs to identify well-known movies from their plots described using emojis. Our evaluation methodology incorporates the use of Chain-of-Thought (CoT) and 4-shot In-context Learning. We randomly select 41 test instances (initially 50 instances, with 9 distracting instances removed, see Appendix D.5) to constitute our test set and arbitrarily designate 4 instances as few-shot contexts. For CoT, we use GPT-4 to generate prompts for each instance in the few-shot context. The model is expected to read the 4-shot in-context examples, generate a thought, and then provide the answer. Our evaluation methodology employs extract string match, i.e. where the output of the model contains the target film name. We set the sampling upper bound times set to be $10^5$.

#### D.4.3 DATE UNDERSTANDING

☝ Date Understanding, a subset of BigBench (Srivastava et al., 2022), is constructed to evaluate the capability of LLMs in comprehending dates, by posing questions related to the date reasoning. For the evaluation of this task, we employ a 4-shot In-context Learning. We randomly sample 47 instances to form the test set (initially 50 instances, with 3 distracting instances removed, see Appendix D.5). We random sample 4 instances from the remaining dataset to serve as in-context examples. We also use extract string match to measure the output from LLMs and set the sampling upper bound times to $10^5$.

| Corpora | Token Portion |
|---|---|
| StarCoder_Python | 0.3 |
| StarCoder_Others | 0.7 |

Table 4: Pre-training corpora used for scaling the Code LLMs (model series 1).

| Corpora | Token Portion |
|---|---|
| StarCoder_Python | 0.15 |
| StarCoder_Others | 0.12 |
| Stack_Overflow | 0.03 |
| Arxiv | 0.05 |
| Pile | 0.65 |

Table 5: Pre-training corpora used for scaling the Code-Text LLMs (model series 2).

#### D.4.4 UNNATURAL IN-CONTEXT LEARNING TASKS

☝ The Unnatural In-context Learning tasks serve as a series of distinctive subtasks within BigBench (Srivastava et al., 2022). These subtasks are designed to assess the models' ability to perform in-context learning where the context sequences are intentionally altered to be likely outside of the training distribution, necessitating the model's attention to unconventional in-context patterns. Some instances of these subtasks are exemplified in Table 6. For each task, 20 instances are randomly sampled to compose the test set, utilizing a 4-shot In-context Learning configuration. Four instances are randomly selected from the remaining dataset to provide context. We use extract string match to measure the output from LLMs and set the sampling upper bound times to $10^5$.

| Task Name | Example |
|---|---|
| Dates | Input: 2015-10-22 Target: *!10!22!2015!* |
| Dates with Unnatural Form | Input: !08!24!1984! Target: *1984-08-24* |
| Dates with Unnatural Content | Input: 96980-49665-10674 Target: *!49665!10674!96980!* |
| Dates with Unnatural Form and Content | Input: !49665!10674!96980! Target: *96980-49665-10674* |
| Identity | Input: a, b, c, d, e Target: *a, b, c, d, e* |
| Reverse Natural Content | Input: t, u, o, b, a Target: *a, b, o, u, t* |
| Reverse to Natural Content | Input: r, o, o, m, s Target: *s, m, o, o, r* |
| 2-digits | Input: 10 - 10 = Target: *20* |

Table 6: Example Tasks in Unnatural In-context Learning Tasks

## D.5 REMOVING DISTRACTING FACTOR IS IMPORTANT WHEN MEASURING TINY PERFORMANCE.

☞ We notice that removing the distracting factor is important when measuring the minor performance gain during scaling. The distracting factor means that a test instance is drastically different from the other test instance in terms of required abilities or evaluation bias. Note that we select the distracting factor based on the observation of test instances, which does not lead to information leakage when predicting the 2.4B model.

For Emoji Movie, some of the movie names are common words, enabling even a modestly sized model to "guess" them correctly based on our assessment criteria: the determination of model correctness is contingent upon the presence of movie names within the model's output. Figure 9 shows that there is no significant association in the pass rates between models of varied scales. In other words, the scaling law does not have much of an impact on model performance for these problems. Consequently, it becomes essential to exclude such distracting factors from consideration. We remove the movie names that are common words identified by the popular toolkit NLTK [5].

For Date Understanding, we omit the following instance shown in Table 7. These instances only require the model to extract the answer from the context and don't require reasoning about the date.

In GPT-4 report (OpenAI, 2023), they split the HumanEval dataset into separate bins with different difficulties and conducted scaling prediction for each bin, thus removing the distraction of easy examples to hard examples.

| Example |
|---|
| Today's meeting is rescheduled to 11 am tomorrow, 10/16/1924. What is the date tomorrow in MM/DD/YYYY? |
| Yesterday was 12/31/1929. Today could not be 12/32/1929 because December has only 31 days. What is the date yesterday in MM/DD/YYYY? |
| Today is 9/7. Jane is watching NFL 2003. What is the date today in MM/DD/YYYY? |

Table 7: Distracting instances in Date Understanding Tasks.

## E  ADDITIONAL EXPERIMENTAL RESULTS

In this section, we display some additional experimental results, including the additional fit curve of dataset level of PASSUNTIL, and the methods of utilizing test loss to assist the instance-level PASSUNTIL estimates.

---

[5] https://www.nltk.org/

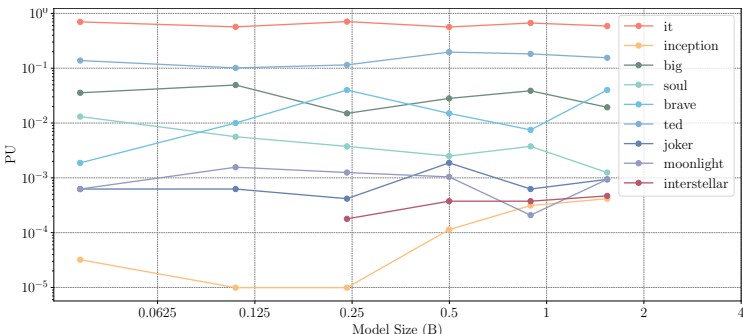

Figure 9: Large and small models have similar PU on these instances (mainly due to randomly sample from the vocabulary space), which creates distracting factors in our experiments.

### E.1 ADDITIONAL DATASET LEVEL PASSUNTIL RESULT.

The performances of series 2 models on HumanEval are represented in Figure 10. This prediction is less accurate compared to series 1. However, with instance level PASSUNTIL, the prediction precision improves.

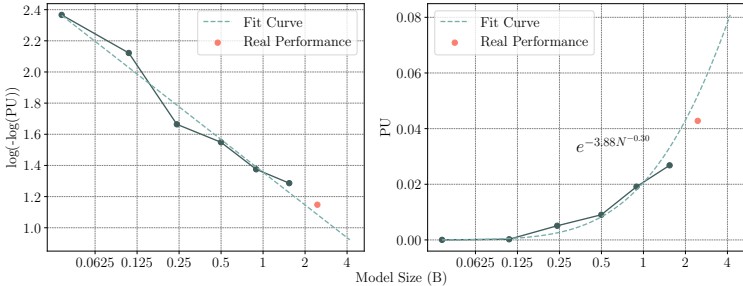

Figure 10: Additional figure on Test Loss Assitted PASSUNTIL Estimate.

## E.2 ESTIMATING PASSUNTIL FROM TEST LOSS

👆 As shown in Figure 11, we propose leveraging test loss on ground truth answers to assist the prediction for "hard samples". For model series 1 and HumanEval task, the linear relationship is found to be $PU \sim 0.22L$. For model series 2 and HumanEval task, the linear relationship is found to be $PU \sim 0.23L$. For model series 2 and Date Understanding task, the linear relationship is found to be $PU \sim 0.96L$. And for model series 2 and Emoji Movie task, the linear relationship is found to be $PU \sim 0.43L$.

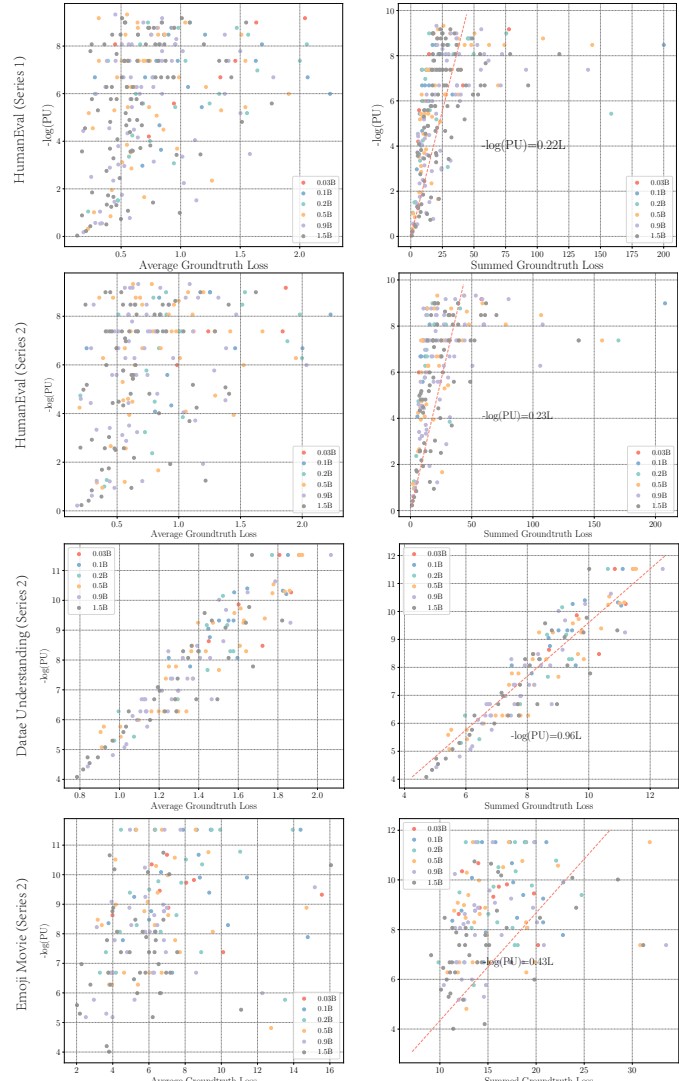

Figure 11: Additional figure on the relation between test loss and PASSUNTIL.

## E.3 MORE RESULTS OF THE UNNATURAL IN-CONTEXT LEARNING TASKS

👆 In Figure 12, we present the scaling curves for the remaining fix sub-tasks of the Unnatural In-context Learning tasks. Notably, the curves in (a), (b), and (c) demonstrate a concave pattern, correlating $\log(\log(-F(N))$ with $\log N$. Specifically, the 2-digits task displays an interesting inverse scaling trend, indicating further investigation to delineate a clearer trend.

Regarding tasks in (d) and (e), we observed that these tasks pose significant challenges for smaller models. Specifically, models with 0.03B and 0.1B parameters failed to achieve non-zero pass rates,

rendering the fit analysis less meaningful. Additionally, for the Reverse to Natural Content task, there's a discernible, albeit slight, sub-scaling law growth trend. This trend may be attributed to the multi-step nature inherent in this task.

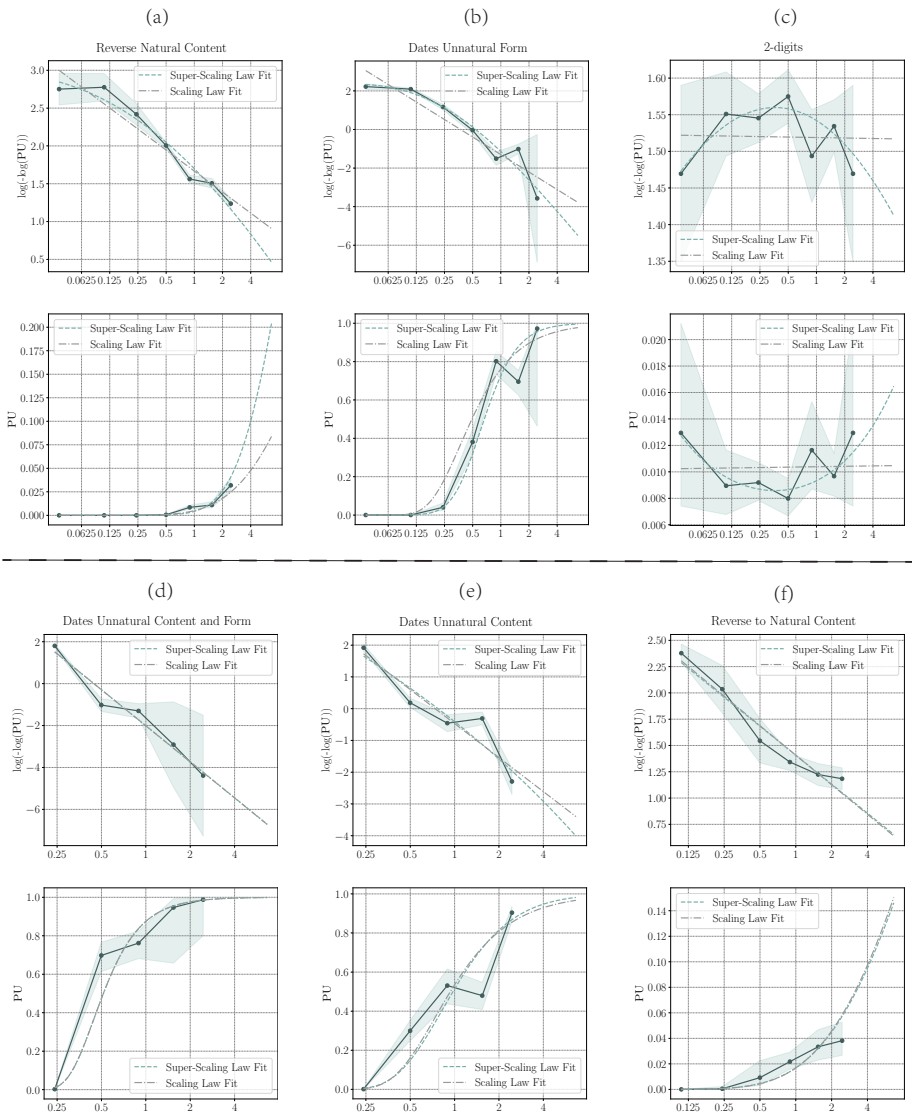

Figure 12: Additional figure on unnatural in-context learning. The grey line shows the scaling law fit, while the green line shows the super-scaling law fit.

### E.4 RESULT OF INDIVIDUAL PASSUNTIL ON MORE SAMPLES

👆 Figure 13 shows more instances of individual PASSUNTIL scaling curves of model series 1 on Humaneval task.

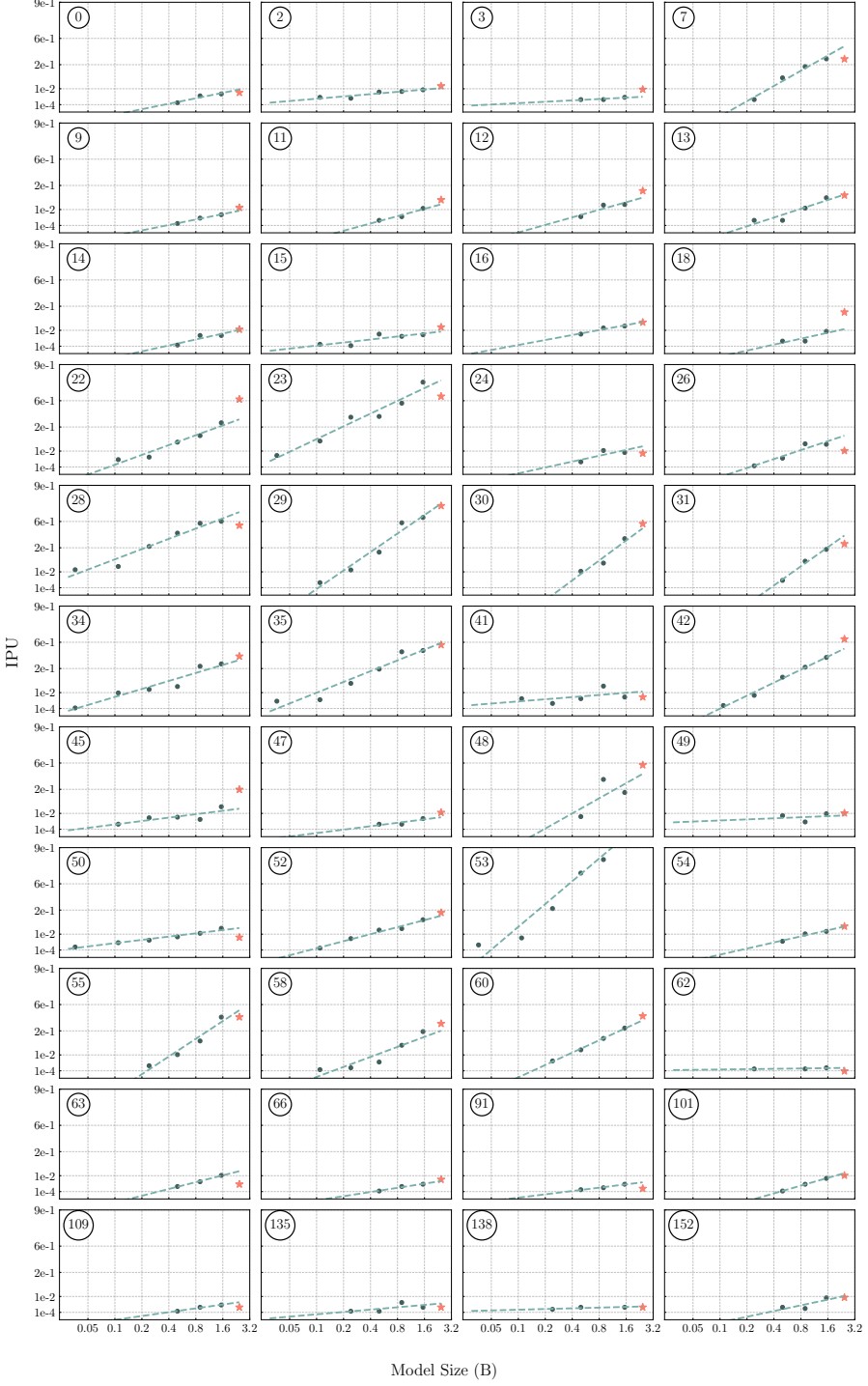

Figure 13: Result of instance-level scaling law fit. The label on the left upper corner of each subplot denotes the index of the sample in the test set[6].

---

[6] https://github.com/openai/human-eval/tree/master/data

