# OpenReview forum: "Predicting Emergent Abilities with Infinite Resolution Evaluation"
_ICLR.cc/2024/Conference — ICLR 2024 poster_

### Official Review · Reviewer_5dKH · 2023-10-13

**Soundness:** 3 good
**Presentation:** 3 good
**Contribution:** 3 good
**Rating:** 6
**Confidence:** 4

**Summary:**

This paper proposes a new approach to "smooth" the emergent abilities. That is, we can sample the model outputs from the logits many times until we get the correct results. The proposed approach can unlock the predictable scaling from emergent abilities. The paper also provides some insights about why chain of thoughts and rethink the relationship between CoT and emergent ability.

**Strengths:**

1) It is an important topic to make the emergent abilities predictable.
2) The insights about the relationship between CoT and emergent ability are interesting. The authors also provide some theoretical evidence about the insights.
3) The proposed approach is easy to implement.

**Weaknesses:**

1) The PathUntil seems to be very expensive in the early stage, because of the low probability of sampling the correct answer.
2) The smoothness of PathUntil highly depends on the output length. For HumanEval it may be okay because the code is simple and short. However, it would be very hard to make it very smooth for the long answers.
3) It would be helpful to provide a more detailed discussion between "ppl on task data" and "passuntil on the task data". I can understand these two are different, but this may be helpful to let more readers to understand the insight of this work.

**Questions:**

1) Can the PassUntil curve be more smooth if we select the "top-K" candidates before sampling? Such as setting a value for top_k in huggingface model.generate API

---

> ### Author Response · Authors · 2023-11-19
>
> Thank you very much for recognizing the significance of our research. Given its theoretical and fundamental nature, which diverages from the typical hot research topics of LLM, we are deeply appreciative of those who can understand the importance of predicting task performance where emergence exists.
>
> ## Weakness 1
>
> Fortunately, the computational cost remains acceptable across model sizes. For large models, we sample a few times, with each generation being slow, and for small language model, we sample a lot of times while each generation is fast. For instance, in our HumanEval experiments, we used a batch size of 200 across 8 GPUs for smaller models, resulting in a processing time of approximately 200 seconds per test example (with a sample number of 10,000).  For 0.03b model, the whole test time for human eval is 40 min. For 2.4b model, it is 2h. Both run in parallelization on 8 A100 GPUs.
>
> ## Weakness 2
>
>  We think that as long as we do not define ''pass'' as generate a sequence exactly the same as the ground truth answer, the smoothness of PassUntil won't be very hard for long generations. Suppose we are now predict the performance on AlpacaEval, who answer is known to be long (> 1000 words). We define ''pass''  as the generation from our model is better than GPT-4, judged by GPT-4. Thus, the probability of generating a ``pass'' answer is not exponentially decrease with the length.
>
> ## Weakness 3
>
> Thank you for highlighting this issue! A significant concern of ours is the common misunderstanding among readers, especially those without access to much pre-training experience, about the difference between 'low perplexity/loss' and 'good task performance.' We are addressing this in our revised paper by adding a detailed discussion on why tracking direct task performance, rather than perplexity/loss, is crucial:
>
> 1. PPL (Perplexity) evaluates aspects different from task metrics. It measures the likelihood of an answer being a natural continuation of a sentence, which is distinct from 'answering a question or do one task successfully'
> 2. PPL cannot accurately inform us of the concrete value of task performance, particularly in practical evaluations.
> 3. While we use loss on ground-truth of the small model to aid in predicting PassUntil, PPL alone is inadequate for predicting performance. This is because, for each sample, the accuracy is binary (0 or 1), which complicates establishing a linear relationship between model performance and PPL
>
> ### Question 3
>
> In our experiments, we set the top-p value to 0.95. Your suggestion to improve the smoothness of the PassUntil curve by further adjusting top-p or top-k is valid. A concern, however, is setting the top-k or top-p value too low, potentially causing the correct answer to be permanently excluded from the candidate pool for small models.

---

> > ### Comment · Reviewer_5dKH · 2023-11-19
> >
> > Thanks for the reply. My concerns have been solved. Please give a more detailed discussion about "what is pass" in your submission. That would be helpful. Also please add the statements in Weakness 3 Response to your work.
> > Since my original score has been positive, I will keep my score. Good Luck!

---

> > > ### Author Response · Authors · 2023-11-19
> > >
> > > Thanks! We are revising the paper according to all reviewers' comments and our statement in response. The revised paper will be uploaded soon.

---

### Official Review · Reviewer_jwFq · 2023-10-29

**Soundness:** 3 good
**Presentation:** 3 good
**Contribution:** 2 fair
**Rating:** 5
**Confidence:** 3

**Summary:**

This paper introduces a novel evaluation strategy, PASSUNTIL, to detect subtle improvements in small models and quantitatively predict the performance. It also presents the task scaling law, which provides insights into the relationship between model size and performance. Additionally, the paper explores emergent abilities through a mathematical analysis of multi-step reasoning and circuit hypotheses. Experimental validation is conducted to support the theoretical analysis.

**Strengths:**

1. Proposed the evaluation strategy "PASSUNTIL" with theoretically infinite resolution, enabling the prediction of task performance and the derivation of the task scaling law.

2. Analyzed emergent abilities using a mathematical definition, challenging prevailing hypotheses and introducing an alternative circuit hypothesis based on theoretical derivations.

3. Conducted experiments to validate the theoretical analysis and provided the first open-source attempt to investigate the predictability of task performance.

**Weaknesses:**

1.	Motivation vs. actual design of the task scaling law:

The paper states that “Despite the predictable decrement in LLM loss, task performance improvements are twisted during scaling” and "First, these works concentrate on training and validation loss metrics, which do not reliably predict task performance." This suggests that the loss metrics may not be highly dependable for predicting task performance. However, the task scaling law design still focuses on the correlation between PU and test loss. This may raise questions about its alignment with the initial motivation because the design of the task scaling law does not go beyond the scope of loss.

2.	Design of the experiment to predict performance:

In the experiments for performance prediction, only the method designed in this paper was implemented, without conducting a comparison with other related methods previously proposed.

3.	The analysis of emergent ability

This work challenges the prevailing hypotheses for emergent ability by employing a redefined metric F(N). However, it should be considered that the choice of metrics, including the new definition of F(N), may potentially impact emergence. How does this work explain the emergence observed in previous works, especially for multi-step reasoning? Does this contradict Theorem 2?

**Questions:**

PU vs. test loss in the task scaling law design: This work seems to give a vague definition of "pass" in PASSUNTIL. Can PU represent all performance metrics of tasks when required?  In the task scaling law, can this relationship between PU and test loss be applicable to all tasks? In the paper "Are emergent abilities of large language models a mirage?" that this work cited, when the task performance metric is changed to Token Edit Distance, the relationship with loss does not align with the formula provided in this work.

This work challenges the prevailing hypotheses for emergent ability by employing a redefined metric F(N). However, it should be considered that the choice of metrics, including the new definition of F(N), may potentially impact emergence. How does this work explain the emergence observed in previous works, especially for multi-step reasoning? Does this contradict Theorem 2?

---

> ### Author Response · Authors · 2023-11-19
>
> Thank you for your insightful review! We believe there may be some misunderstandings that have led to the questions raised.
>
> ## Weakness 1
>
> Sorry for the confusion, reviewer 5dKH weakness-3 indicates that it's our fault that we do not make the point clear. Let us make the point clearer.
> - It's important to distinguish between '**loss is not predictive of task performance**' and '**loss can help predict task performance.**' The former suggests that loss is a not sufficient statistic for estimating task performance without other measurement, while the latter indicates that loss is one of useful factors in improving prediction accuracy. In our paper, we clearly verify both statements. Without utilizing the PassUntil method, one cannot deduce actual performance (accuracy) solely from loss values. For example, a loss of 1.0 does not directly translate to an accuracy of 0.2 for a task. And **actual performance must be empirically measured**. Furthermore, as shown in our pre-revision Figure 4, the loss of an individual sample does not have a one-to-one correlation with PassUntil results, much less with discrete accuracy.
> - However, loss does provide useful information. Once we measure PassUntil across a large sample set, we can establish a statistical relationship between loss and PassUntil (not possible if we only rely on loss data). This relationship can enhance our prediction accuracy.
> - The incorporation of loss for improved predictions is driven by practical considerations, such as limited computational resources, rather than being a necessity. Figure 3 demonstrates that even without loss data, we can accurately predict task performance. Imagine a scenario where we can measure every sample with sufficient resolution to ensure each is passed at least once; in such a case, loss data would not be necessary.
>
> ## Weakness 2
>
> As the first open-sourced project attempting to predict the scaling curve of task performance, we are not aware of any other existing methods capable of such prediction. If the reviewer is aware of related methods, we would appreciate references to enhance our understanding.
>
> ## Weakness 3
>
> We have addressed the confusion regarding the Definition of Emergence in [our global response]（https://openreview.net/forum?id=lDbjooxLkD&noteId=qUjpV9mmJv）. Please let us know if there are any further points of confusion.
>
> ## Question 1
>
> - Our methodology currently applies only to generative tasks where the success rate is zero for random generation. This limitation is outlined in Section 4.1's comments and the Limitations section in the Appendix. However, it appears that most tasks inherently fit this generative model with some pass criteria if the test format is adapted accordingly. We will expand on these limitations in the body of our revised manuscript.
> - You mentioned 'Token Edit Distance,' which is predominantly used in multiple-choice tasks. While this metric offers more continuity than accuracy for such tasks, establishing a direct link between 'Token Edit Distance' and accuracy remains challenging. Therefore, we have deferred the exploration of these types of tasks to subsequent work.

---

### Official Review · Reviewer_v7YM · 2023-10-29

**Soundness:** 4 excellent
**Presentation:** 4 excellent
**Contribution:** 4 excellent
**Rating:** 8
**Confidence:** 5

**Summary:**

Previous work pointed out that limited resolution may yield biased estimates of LLM performance. This paper proposes sampling until a non-zero number of success is achieved, then shows doing so allows for better estimation of small quantities.

**Strengths:**

Overall, I think this is a good paper. I think it is well motivated, thorough and insightful. I would be happy to increase my score if a number of modifications are made (or if the authors tell me why I'm mistaken!).

**Weaknesses:**

Here, I order my feedback in sequential order based on moving through the paper top to bottom.

> We hypothesize that the perceived discontinuity from trivial to excellent performance might stem from limited evaluation resolution. By employing a more nuanced resolution, one could potentially uncover the scaling law for tasks. Our hypothesis diverges significantly from that of Schaeffer et al. (2023),

I think this misstates Schaeffer et al. (2023). Their abstract states, “we provide evidence that alleged emergent abilities evaporate with different metrics or _with better statistics_”. The “better statistics” refers to increased resolution. Specifically, their Figure 4 shows that increasing resolution by sampling significant amounts of additional data improves predictability on the original metric i.e. Accuracy. One of Schaeffer et al. (2023) et al.'s key takeaways is that predicting capabilities with scaling can, in certain cases, demand significantly higher resolution.

>  PASSUNTIL deploys extensive random sampling in the decoding phase (e.g., 105 sampling times), and evaluates each sampling result until any generation passes the target test metric. Therefore, this evaluation strategy has infinite measurement resolution as long as computational resources are not bounded.

Assuming unbounded computational resources is not a trivial assumption to make, and indeed, in the experiments, the authors do not sample as dictated by their procedure. Rather, the authors set a finite upper bound on the number of samples. Consequently, I am minorly concerned with their story and with their results since the Pass Until (PU) score they discuss is **not** the PU score they numerically compute. I think they should use PU as defined or offer a correction if a limited compute budget exists.

> Theorem 1.

The authors note that the failure time $f = K - r $ follows a negative binomial distribution. I have two comments.

1. Wikipedia seems to suggest that the author's estimator $PU := r/\mathbb{E}[K]$ is a *biased* estimate (https://en.wikipedia.org/wiki/Negative_binomial_distribution#Maximum_likelihood_estimation), directly contradicting Theorem 1. The citation given is Haldane Biometrika 1945 "On a method of estimating frequencies" and a quick Google search discovers many Math.StackExchange & Cross Validate.StackExchange posts backing up Wikipedia. Could the authors please double check? I suspect the step in their derivation of dividing by $\mathbb{E}[K]$ may be incorrect, but I don't have time to investigate. My guess is that if the estimator is defined as $\hat{P}(s) := \frac{r}{K}$, then its expectation $\mathbb{E}[\hat{P}(s)] := \mathbb{E}[\frac{r}{K}] \neq \frac{r}{\mathbb{E}[K]}$, as the authors claim.

2. When $r$ is known, the minimum variance unbiased estimator (MVUE) for a Negative Binmoial is known to be $\frac{r-1}{K-1}$ (where K is the total number of trials to get $r$ successes, in the authors' notation; https://en.wikipedia.org/wiki/Negative_binomial_distribution#MVUE_for_p). I am curious to know why the authors chose to not use this estimator? If the authors have a compelling reason, I would be happy to listen. If the authors do not have a compelling reason, I would strongly recommend changing to this MVUE and recomputing their results; doing so should not require resampling from the models. Defining PassUntil as this MVUE would also salvage Theorem 1.

> Table 1

I think this Table could be much more easily understood as a Figure. Assuming you're using Python, I recommend doing the following in Matplotlib & Seaborn: Create two axes, one for Emoji Movie & Date Understanding. Then, the x axis of both matplotlib axes should be number of instances passed (from 0 to 41). The y axis should be the 4 models, and the 4 sampling options (BS, RS-1, RS-100, RS-10000) should be the hues.

> Equation 4

Two comments here:

1. I believe this equation is incorrect. Equation (3) is a product from $i=1$ to $i=|y|$, and Equation 4 correctly converts that product into a sum, but the right hand side has no dependence on $|y|$. Your Equation 4 states that generating a length 1 output and generating a length $10^9$ output have the same PU scaling. Rather, I believe you should have $\exp(-c |y| N^{-\alpha}).

2. Once corrected, your Equation 4 is identical to Schaeffer et al.’s $Accuracy(N)$ equation (Page 4) (modulo the slightly different definitions of the scaling law form). It would be good to acknowledge this.

> Table 2: Prediction of our framework

Here, I also recommend converting the table to a figure. Also, where are your confidence intervals or standard errors?

> Figure 6: Scaling curve for task “Dates” and “Identity”.

At a granular level, where are the error bars? You should be able to repeat this experiment by bootstrapping which 20 questions are in the test set and which are the in-context examples.

At a higher level, I'm a bit skeptical about this contribution for a few meta-reasons.

1. Previous emergent papers didn't emphasize this BIG-Bench task (Unnatural In-Context Learning) and to be honest, I haven't heard it mentioned as an emergent ability.
2. The task specifies 8 subtasks (https://github.com/google/BIG-bench/tree/main/bigbench/benchmark_tasks/unnatural_in_context_learning#subtasks) but this manuscript only considers 4 and I don't know why those 4 were chosen.
3. The curves are fit with no held-out data for testing generalization of the fit.

Put together, I'm unsure whether these specific tasks might have been cherry picked. BIG-Bench has >100 tasks, so I'd guess any pattern can be found if one looks for it. That doesn't mean the pattern is a general phenomenon; rather, by random chance alone, some curves will look concave or convex with respect to log N.

Lastly, by limiting the number of samples ($10^5$), I believe the PU estimator is no longer unbiased. The whole promise of PassUntil is that one can keep sampling so long as necessary. Is this correct? If so, how can we correct for the finitely many samples?

> if F (N ) is a concave function of log N , then the task obeys super-scaling law growth, which is defined as “emergent”

I have two thoughts here. On one hand, I feel like "Emergence Still Exists But Becomes Continuous." is an oxymoron based on the definition offered by Wei et al. 2022, and that one shouldn't attempt to overwrite an established definition. But on the other hand, a better definition is a welcome invitation. Perhaps it would be better to more clearly title this section "An Alternative Definition of Emergence" or "A Quantitative Definition of Emergence" to make clear that the existing definition is being more precisely defined/refined.

> Theorem 2. Suppose each reasoning step’s success rate, measured by PASSUNTIL obeys the scaling law growth, then the multi-step success rate follows the sub-scaling law growth.

Disclaimer: Empirically, this seems wrong to me. Looking at Figure 7 (right), subscaling behavior is _not_ what multi-step success task curves look like.  I think the problem is the assumption: "Suppose each reasoning step’s success rate, measured by PASSUNTIL obeys the scaling law growth." What I've observed is that LLMs exhibit non-trivial conditional dependencies such that if the first few tokens are correct, subsequent tokens are more likely to be correct, and vice versa. I'm not saying the math is incorrect. I'm saying that this assumption isn't a good fit to reality. This is also visible in Schaeffer et al. 2023's Figure 4: the independence assumption results in geometric decay, but LLMs don't decay this quickly.

> Theorem 3. In this scenario, the success rate of the task is the majority voting of these circuits

This statement is unclear. I'm willing to buy that multiple circuits exist, but why should the decision-making be via majority vote. Is this an assumption? If so, I recommend changing the language to "In this scenario, *if* the success rate of the task is the majority
voting of these circuits." But if not, then could the authors please clarify why decision-making is via majority vote?


> set the sampling upper bound times to $10^5$

In my experience, this takes a significant compute budget. Could the authors please quantify how long such evaluations take in compute, time and money?

## Possibly Unacknowledged Limitation

Some LLM evaluation processes don't involve sampling. This includes non-generative tasks e.g., MMLU or greedy decoding aka temperature=0. In such evaluation processes, does PassUntil make sense? And if not, hopefully the authors can add a limitation section.

## Suggested Alternative Title

I won't take this into account for the review, but reading the title "Unlock Predictable Scaling from Emergent Abilities," I don't know if "from" is the right preposition. I'd recommend a slightly different title like "Predicting Emergent Abilities of LLMs with Infinite Resolution Evaluations". I think that's more clear and also more exciting.

**Questions:**

1. Why did you need to train your own models? What's wrong with using existing models e.g. Llama, Mistral, Pythia? Using existing models so seems much cheaper.

2. [Disclaimer: I'm not affiliated with the following paper] What do y’all think of Arora & Goyal et al. 2023’s A Theory for Emergence of Complex Skills in Language Models https://arxiv.org/abs/2307.15936? It seems to disagree with your Theorem 2, if I understand correctly. It'd be good to know what the relationship is between these.

---

> ### Author Response · Authors · 2023-11-18
> **Response (Part 1)**
>
> First and foremost, we want to express our heartfelt thanks to Reviewer v7YM. You are not just a professional, meticulous, and thoughtful reviewer, but also a sage brimming with insight and patience. Your comments are truly inspiring. Thank you reviewing our paper and we wish you every success and happiness in life. Even though eventually we might not convince you with our following answers, we would like to thank you for improving the quality of our paper.
>
> The updated version will be uploaded after we answer all reviewers' questions.
>
> ### Weakness 1
> - We appreciate your observation regarding the misattribution, and we apologize for any confusion caused. Our emphasis on distinguishing our work from Schaeffer et al. (2023) arose from our initial feedback after share our work with colleagues, where some perceived our approach as another version of ‘smooth metrics’, which led to doubts about the practicality of our method for predicting actual metrics of interest. Our contrast was specifically directed only at this aspect of their work, not their other conclusions, which we believe may have led to misunderstandings among readers.
>
> - We also acknowledge our oversight in not sufficiently recognizing Schaeffer et al.'s discussion on resolution. In our revised paper, we will duly credit their insights on this aspect and state that our approach delves deeper into enhancing resolution based on the observation of Schaeffer et al. In fact, unlike Schaeffer et al., who increase resolution by expanding test data (an impractical approach for real test sets), our method improves resolution without the need for additional test data. This innovation allows for the prediction of more challenging tasks and represents the first open-sourced success to use this approach for predictable scaling.
>
> ### Weakness 2, 3
> Regarding the second and third questions about the PassUntil method, we will address them together:
> - We apologize for the confusion caused by our erroneous use of the term 'unbiased' estimate. During our method's design, we were aware of the 'biased' statement on Wikipedia. Our initial solution was to move the expectation to the denominator. With this remedy, the math itself is correct: $E[K] - r = E[f] = r(1-P(s))/P(s) = r/P(s)$, thus $P(s) = r/E[K]$.  Practically, if we repeat experiments, we first take expection of $K$, then get $P(s)$.  However, upon further reflection, we realize that this make $r/E[K]$ as a constant, rather than a random variable for estimating $P(s)$, makes 'unbiased estimate' an inappropriate term.
> - Regarding the MVUE estimate you proposed, we find it theoretically sound but encountered an issue when setting $r$ to 1, resulting in an MVUE of $0$ for $P(s)$ as calculated by $(r−1)/(K−1)$. **We seek your guidance for a suitable MVUE when r=1**. (Negative Binomial Distribution itself doesn’t require $r>1$). Once that is solved, we would be happy to migrate the whole method to (r-1)/(K-1).
> - Our current solution for the statement in the paper involves **shifting from 'unbiased estimate' to 'maximum likelihood estimate' (MLE).** The MLE, particularly $r/K$, suffices for most scenarios, including ours.
> - Notably, fixing the sample number $K$ to a large constant value and make $r$ a random variable also renders $r/K$ an effective MLE. This might partly resolves your concern about the case of PU score when computation limit exists. i.e., if we set all runs to sample for a fixed times $K$, and observe the number of success $r$, the MLE we get is still $r/K$. I wonder if this explanation might be satisfying to you.
>
> ### Weakness 4
> - Please see the changes in updated version.
>
> ### Weakness 5
> - In our equation $c$ is set to $c = \sum_i c_i $ (we make that clear in our revision). So the equation is not incorrect. We add an acknowledge of Schaeffer et al.’s contribution besides the original referenced OpenAI citation.
>
> ### Weakness 6
> - Please see the changes in updated version.
>
> [To be continued]

---

> ### Author Response · Authors · 2023-11-18
> **Response (Part 2)**
>
> ### Weakness 7
> We do not cherry-pick the results. As this research represents an initial attempt into predictable scaling, we selected tasks based on specific criteria for simplicity:
> 1. The tasks are not of a multiple-choice grade. This exclusion eliminated 153 tasks. Our investigation into the predictable scaling of multiple-choice tasks is under-progress  for a future publication.
> 2. The answers are longer than a single word, such as ASCII Word Recognition, as they do not align with our sampling procedure.
> 3. The tasks are solvable by our largest model, the 2.4B, under natural metrics (for the sake of verification of prediction). This criterion ruled out tasks like chess_state_tracking, hindi_question_answering etc.
>
> Based on these criteria, we identified the 'Unnatural In-Context Learning' task from BigBench as most suitable for our current stage of research.  As for the task has 8 tasks, we will add the rest 4 subtasks in our revision.
>
> It's important to clarify that our primary goal is on making precise predictions based on our scaling experiments. Hence, the breadth of task variety is not our immediate priority. However, we are committed to continuing this line of research and aim to achieve predictable scaling across a broader range of tasks in future studies.
>
> ### Weakness 8
> Yes, we highly agree. After we have submitted our paper, we also thought that the word ''emergent'' has been used to describe the sudden rise in performance and it's not very appropriate to re-define it and causing misunderstandings. We consider to name the phenominon of ''super-scaling law behavior'' as ''**accelerated emergence**'' in the revised version. With that definition, emergence can still be empirically understood as ''exponential-like growth'' in task performance. And among which,  a part of emergent task still obeys the task scaling law. While we are discovering the new case of emergence that grows faster than ``scaling law-induced emergence''.
>
> ### Weakness 9
> We respectfully question your assertion that subscaling behavior does not resemble multi-step success task curves. Prior to our mathematical categorization, the performance of all emergent tasks exhibited characteristics akin to 'exponential growth',  especially when the slope in logarithmic space was significant. They are not easy to distinguish unless you view it from a log space and with mathematical characterization. Additionally, we observed that LLM performance does indeed decrease geometrically. This observation aligns with Schaeffer et al.'s statement `... as the target string length increases, ... geometrically ... ''. And this geometrical growth emerges when each step is governed by a consistent scaling law coefficient, resulting in an expression like $\exp(|y|N^\alpha) = \exp(N^\alpha)^{|y|} $.  We wonder whether the reviewer could back the interesting observation with some reference? Such insights could be invaluable for our further investigation.
>
> ## Weakness 10
> - Yes, it’s an assumption, we will make this clear as you suggested.
>
> ## Weakness 11
> - Actually, the trade-off between actual sampling numbers and time cost for each sample is interesting. For small models, they need more sampling iterations, but each sample is super fast. For large models, their generation is slow, but they success within a few samples. For 0.03b model, the whole test time for HumanEval is 40 min. For 2.4b model, it is 2h.  All evaluations run parallelly on 8 A100 GPUs.
>
> ## Limitation 1
>  We have mentioned it in limitation section. We will make it clearer in revision.
>
> ## Title
> We buy it! Thanks a lot for the excellent title!
>
> [To be continued]

---

> ### Author Response · Authors · 2023-11-18
> **Response (Part 3)**
>
> ## Question 1
> - We do need to train our own model, since the prior goal of this work is to make precise prediction on the actual performance of our trained model. For Pythia and OPT, they do not faithfully obey the loss-scaling law. For example, you can see the loss visualization of OPT checkpoints in [this paper](https://arxiv.org/pdf/2212.09803.pdf) ’s Figure 1.  Clearly they do not obey loss scaling law. Therefore, we only use them for the qualitative study in Table 1 (before revision). For llama and mistral, they only open opensourced less or equal to four checkpoints of different sizes, which is not enough for fitting a scaling curve for making predictions about the largest size checkpoint. And their fit to scaling law is not satisfying as well. **Using their checkpoints, we can make valuable observation, but we can not make predictions**. We verify our models follow loss scaling law strictly in our Figure 2, which provides a solid foundation to task performance prediction.  We will open-source all our checkpoints to facilitate prediction scaling in the community.
>
> ## Question 2
> It is not contradict to our theoreom 2. The misunderstanding is indeed caused by our attempt to overwrite an exstablished concept. Multiple step reasoning, or complex skills for solving a task, will make the task scaling law's slope steeper in the log space, which results in a steeper  ''exponential form'' in the task performance. However, our defined ''accelerated emergence'' (originally named as emergence), is **not about the speed**, but about the **acceleration**. The acceleration is only distinguishable after the mathematical form has been established. And we actually means that ''multistep reasoning'' does not lead to super-scaling law behavior, a.k.a., accelerated emergence. We will make it cristal clear in the revision.
>
>
> To sum up, we owe many thanks to Reviewer v7YM, who has made concrete contribution to the quality improvment of our work. His/Her professional knowledge, commitment, carefulness, and patience deeply touches us, making us believe ICLR's review quality is at the apex of AI conference. We will add Reviewer v7YM to our acknowledgements despite that we don't have the reviewer's name.

---

> > ### Author Response · Authors · 2023-11-23
> >
> > ```
> > Table 2: Prediction of our framework
> >
> > Here, I also recommend converting the table to a figure. Also, where are your confidence intervals or standard errors?
> > ```
> > Thank you for your suggestion regarding Table 2. After careful consideration, we have decided not to convert it into a figure. The purpose of Table 2 is to present concrete values that illustrate the numerical differences between our framework's predictions and actual values. This specificity is best conveyed in a tabular format. For a visual representation of the scaling curve's estimates, we direct your attention to Figure 6, which we believe may align with your request for a figure.
> >
> > Regarding the inclusion of confidence intervals or standard errors, these metrics are less relevant in the context of our methodology. Our framework integrates an instance-level fit from the PassUntil approach with the assistance of a loss function. This combination complicates the computation of meaningful confidence intervals or standard errors, as it moves beyond traditional statistical modeling where these measures are most applicable. We believe that the robustness of our results is adequately demonstrated through the methods and results described in our study.
> >
> > We appreciate your insightful feedback and **have thoroughly addressed all other concerns in our revised manuscript**. Your input has been invaluable in enhancing the quality and clarity of our work. Thank you once again for your detailed review and constructive suggestions.

---

### Official Review · Reviewer_BHGc · 2023-10-30

**Soundness:** 2 fair
**Presentation:** 3 good
**Contribution:** 3 good
**Rating:** 5
**Confidence:** 3

**Summary:**

The paper aims to understand the scaling properties of large language models (LLMs) and discover a task scaling law to predict task performance. The authors introduce an evaluation strategy called PASSUNTIL, which utilizes massive sampling in the decoding phase and has theoretically infinite resolution. They conduct a quantitative investigation into the scaling law of task performance, leading to two main findings: (1) Task performances are predictable with PASSUNTIL; (2) The emergent abilities do exist and can be characterized by super-scaling law.

**Strengths:**

1. This paper presents a pioneering open-source effort to predict task performance in large language models, aiming to significantly contribute to and encourage future research in this area.

2. The authors introduce PASSUNTIL, a novel evaluation strategy that appears to provide a more equitable assessment compared to existing metrics, showcasing their innovative approach to addressing the challenges in the field.

3. The proposed scaling laws demonstrate a strong fit with the data across several datasets, highlighting the effectiveness and potential applicability of the authors' findings in various contexts.

**Weaknesses:**

1. The derivation of Equation (3) appears to have some discrepancies. (Please refer to Question 1)

2. The number of evaluation datasets used in this study is somewhat limited in comparison to previous works on scaling laws. (Please see Question 2)

3. The task scaling law seems somewhat arbitrary, as some tasks require standard scaling laws while others necessitate super scaling laws. (Please refer to Question 3)

Overall, this paper serves as a valuable starting point, but it lacks a comprehensive investigation.

**Questions:**

1. Scaling laws are generally in an average sense over a dataset but not for individual tokens. For example, in a single generation, an error incurred in earlier tokens is likely to cause errors in subsequent tokens.

2. This paper evaluates the "emoji," "date," and "identity" tasks. However, the number of evaluation datasets seems relatively small compared to existing works like Wei et al., 2022a. As a result, it is uncertain whether the conclusions drawn in this paper are broadly applicable. Could you please discuss the generalizability of your findings?

3. The scaling laws in Equation (4) are for general tasks, irrespective of whether they involve multi-step reasoning. Ideally, they should be applicable to all tasks. However, Figure 6 demonstrates that it does not work well for multi-step reasoning tasks, and the authors propose another super scaling law to address this issue. It makes this paper a bit ad-hoc.

4. Theorem 2 is based on the premise that emergent abilities are induced by multi-step reasoning. However, Appendix E of Wei et al., 2022a includes numerous non-reasoning tasks that exhibit emergent abilities.

5. In the evaluation of PU, both beam search and random sampling decoding are utilized. Does the temperature setting influence the performance, and if so, how is this factor taken into account?

---

> ### Author Response · Authors · 2023-11-15
> **Response to regarding the four questions**
>
> **Question 1**
> - A1). The example provided does not account for the discrepancies noted. In our methodology, we estimate the probability of generating the **correct** answer (otherwise, why should we predict an answer’s probability which contains error itself?). Since the answer is correct, we assume it to be a natural continuation consistent with the training distribution. Therefore, each token is conditioned on the preceding correct (or natural) token, without any internal errors. This allows us to apply the scaling law to each token to calculate the total probability of the answer.
> - A2). While scaling laws are typically applied in an average sense, this does not preclude their adaptation to individual token levels. Theoretically, a scaling law implies linearity in the logarithmic space of model parameters/compute and the logarithmic space of loss. If single-token losses were not linear in this log space (There must be more random noise at instance-level, though), it would be challenging to understand the observed linearity on average. This is because numerous non-linear structures would dominate the scaling process, leading to non-linearity in aggregate. To provide a more rigorous analysis, prompted by your question, we are examining the scaling relationship of single-token loss. We do linear regression `log(loss) ~ log(modelsize)` on each token of  HumanEval's ground-truth solution. And we count the R^2's distribution. which is
>
> |     R^2    | 0-0.2 | 0.2-0.4 | 0.4-0.6 | 0.6-0.8 | 0.8-1.0 |
> |---------|-------|---------|---------|---------|---------|
> | Frequency | 0.17  | 0.12    | 0.15    | 0.25    | 0.29    |
>
> So, at token level, randomness indeed dominate the loss, but there are still a considerable tokens obeying scaling law. Therefore, we think the assumption is reasonable at the deduction stage. A careful dive into the R^2 interval provide us the observationn that R^2=0-0.2 contains the tokens that are either too easy or too hard to predict.
>
>
> **Question2**
>
> Our choice of evaluation tasks can be attributed to two primary considerations:
> - 1. Our method is currently designed for tasks where a random guess would typically result in a zero score, particularly in smaller models. This limitation is detailed in Section 4.1, Comment 2.  However this does not compromise the general applicability of our approach. Generative tasks are inherently versatile, making our method suitable for a variety of applications. Moreover, other metrics could be build upon our method. For example, a possible solution for multiple choice grade metric is to track each choice’s PassUntil with task scaling law or just track the Brier Scores.
> - 2. Another consideration of our task selection from practical perspective is that the task can be evaluated on a simple pass/fail basis. While our method is capable of processing generative tasks that requiring complex human evaluation, these would necessitate the support of an answer-judging model, which will complicate our initial attempt for predictable scaling.
>
> We must point out that **observing** scaling behavior (what Wei et al.2022a do) is easier than **predicting** it.  Despite that, we are currently expanding our scope of prediction!
>
> **Question3**
>
> Equation (4) was build upon very strong assumption (each token’s scaling speed is the same, and there is one circuit to get the answer passed in terms of neural circuit), without which we can not achieve any meaningful formulation of the scaling properties of task performance. We can see there are practical tasks (HumanEval, Date Understanding, etc.) do show such scaling properties.  However, since the assumption is so strong, we are not surprised to see the cases (in figure 6) that does not obey the assumption.  In fact, the whole section 6 is to discuss the more general case of task scaling law. We can see that if each token’s scaling speed is not the same, it must induce a sub-scaling law growth instead of super-scaling law growth. Which is not what we observed in experiments. So we propose another hypothesis based on neural circuits. The methodology we have adopted is not arbitrary or ad-hoc. Instead, it represents a systematic and gradual approach to addressing a complex scientific problem.
>
> **Question4**
>
> That ``emergent abilities are induced by multi-step reasoning'' is not an assumption made by this work, it is made by Wei et al.2022a in Section 5.1. We are refuting such assumption by formalizing it into a theorem to check it correctness. The non-reasoning tasks in Appendix E of Wei et al. indeed support our view against multi-step reasoning assumption.
>
> **Question 5**
> Temperature affects the performance. However, currently we didn’t take into account the variation of temperature during model scaling, we use the same temperature for the small LLMs as the large LLMs, which is a natural choice considering the current prevalent use of random sampling in LLMs' generation.

---

### Author Response · Authors · 2023-11-15

Thank you for all the insightful review! As the first team to develop an open-source solution for predictable scaling, our framework was built from scratch. This might have led to some confusion. We think that any discussion on our methodology is very valuable for the community to develop predictable scaling techiniques!

We appreciate the suggestions regarding the generality of our test set. **We are currently conducting further experiments** to address these points. However, we do think that we should be patient in the pursuit of predictable scaling, since even OpenAI only reports one positive result on predictable scaling, i.e., HumanEval, and we are able to achieve that prediction. We hope to have the further results ready before the discussion concludes.

In the meantime, we will address the other questions to facilitate the start of our discussion.

---

### Author Response · Authors · 2023-11-19
**General Comments on ''Emergence'' and Theorem 2**

We apologize for any confusion caused by our ‘’new definition of emergence’’.

Initially, our paper was premised on the idea that task scaling laws $\exp(-cN^{-\alpha}) $ could be derived from loss scaling laws $ L = L_0 + cN^{-\alpha} $. And since loss scaling is not considered emergent, we posited that task scaling laws should not be classified as 'emergent' either, despite their natural exponential form (as depicted in Figure 7). Based on this perspective, we redefined 'emergence' as a scaling trend without a constant scaling factor $\alpha$, thus not fitting any $\exp(-cN^{-\alpha})$ model.

However, after reflection and discussions with Reviewer v7YM, we recognize the importance of ''not overwrite an established concept'' to avoid causing a lot of misunderstanding among the field. Therefore, we will **retain the original definition of 'emergence' as simply (and qualitatively) 'the sudden increase in task performance'**, even when such an increase conforms to the task scaling law $ \exp(-cN^{-\alpha}) $ and has good predictability.

Instead, we propose the term **'accelerated emergence' from a quantitative perspective** for tasks whose performance scaling cannot be fitted by any standard task scaling law, indicating they follow a 'super-scaling law curve.' This term describes tasks that are currently 'unpredictable' and warrant special attention due to their complexity and unpredictability.

This redefined concept of 'accelerated emergence' should clear up several misunderstandings. For instance, our Theorem 2, which states that 'the multi-step success rate follows sub-scaling law growth,' does not contradict hypotheses suggesting that multi-step reasoning leads to the ''original observation of emergence''. In many observations, emergence is simply a 'sudden breakthrough in performance.' And both sub-scaling law growth and scaling law growth can result in such a breakthrough when viewed discretely (as shown in Figure 7). Our argument for Theorem 2 is that multi-step reasoning does not lead to 'accelerated emergence' or super-scaling law growth.

---

> ### Comment · Reviewer_v7YM · 2023-11-20
> **Response to General Comments on ''Emergence'' and Theorem 2**
>
> > Instead, we propose the term 'accelerated emergence' from a quantitative perspective
>
> That sounds very good. Thank you for thinking this through carefully!

---

> > ### Author Response · Authors · 2023-11-20
> >
> > You're welcome! It's our duty to make the point clear.

---

### Author Response · Authors · 2023-11-23
**Summarization on changes made in the Revision PDF**

We have address all point from the reviewers in the revised version

[Reviewer v7YM]
1. We change the title.
2. We add attributions of Schaeffer et al. (2023) to both introduction and Equation (4), along with thorough modifications to accommadate the change.
3. We change the ''unbiased estimate'' statement to ``maximum likelihood estimate''.
4. We change the ''emergence'' that we originally propose to redefine into ``accelerated emergence'', with thorough modifications to accommadate the change.
5. We add hues to the scaling law figure in Section 6, and Appendix E.3
6. We add the result of the rest of 4 subtasks in unnatural in-context learning tasks, and provide their analysis in Appendix E.3
7. We change the ''Comments section'' into two ''Necessity'' and ''Limitation'' section in 4.1
8. We make the Thereom 3's assumption clearer by changing ''In this scenario, ...'' to ''suppose ...''
9. We change the Table 1 to a figure

To sum up, we have tried to address all Reviewer v7YM except for one that we explained to his part of review.

[reviewer 5dKH]
10. We add a discussion in the footnote in Page 4 to illustrate what is ''pass''.

[reviewer jwFq]
11. We add a discussion to discuss why using loss to assist prediction is not contradict to our motivation of proposing PassUntil in Appendix A.2.

In the revised version, revisions are marked in red. Please note that additional modifications were made for consistency throughout the paper, but these are not highlighted in red to avoid excessive marking.

---

### Meta-Review · Area_Chair_UdGf · 2023-12-11

**Metareview:**

This paper studies task scaling law. Typically, scaling laws focus on the optimization loss but the task specific scaling laws have not been examined carefully. This paper makes an interesting finding: that small models show critical and consistent task performance improvements that are not captured by conventional evaluation strategies mainly due to insufficient measurement resolution. They further propose a evaluation strategy to address this issue and demonstrate it empirically.

**Justification For Why Not Higher Score:**

The evaluation scope of the paper is somewhat limited. Also, more discussion regarding points raised by Reviewer BHGc needs to be added to be the paper.

**Justification For Why Not Lower Score:**

The paper makes some interesting observations and can be valuable to be known to the wider community.

---

### Decision · Program_Chairs · 2024-01-16

Accept (poster)